# Unraveling radiation damage and healing mechanisms in halide perovskites using energy-tuned dual irradiation dosing

Ahmad R. Kirmani [1,2] ✉, Todd A. Byers [3], Zhenyi Ni [4], Kaitlyn VanSant[1,5], Darshpreet K. Saini [3], Rebecca Scheidt[1], Xiaopeng Zheng[1], Tatchen Buh Kum[2], Ian R. Sellers [6,7], Lyndsey McMillon-Brown [5], Jinsong Huang [4], Bibhudutta Rout [3] & Joseph M. Luther [1] ✉

Perovskite photovoltaics have been shown to recover, or heal, after radiation damage. Here, we deconvolve the effects of radiation based on different energy loss mechanisms from incident protons which induce defects or can promote efficiency recovery. We design a dual dose experiment first exposing devices to low-energy protons efficient in creating atomic displacements. Devices are then irradiated with high-energy protons that interact differently. Correlated with modeling, high-energy protons (with increased ionizing energy loss component) effectively anneal the initial radiation damage, and recover the device efficiency, thus directly detailing the different interactions of irradiation. We relate these differences to the energy loss (ionization or non-ionization) using simulation. Dual dose experiments provide insight into understanding the radiation response of perovskite solar cells and highlight that radiation-matter interactions in soft lattice materials are distinct from conventional semiconductors. These results present electronic ionization as a unique handle to remedying defects and trap states in perovskites.

Future generation low-cost and lightweight photovoltaic (PV) technologies for powering space vehicles and satellites should be tolerant to the space environment including radiation, atomic oxygen, thermal cycling, and vacuum[1]. Solar cells based on halide perovskite semiconductors appear to meet most of these criteria and are now beginning to be explored for future space applications including space-based solar power and as solar panels to power satellites in Earth orbits[2]. Initial reports suggest unique radiation tolerance of perovskite solar cells (PSCs), superior to the conventional PV technologies based on Silicon and III-V semiconductors currently used in space[3–9].

Protons generated by low-energy accelerators provide an effective source for assessing radiation hardness of PSCs given their efficiency in creating atomic displacements, and their ability to mimic the radiation spectrum in Earth orbits[10]. However, radiation in space is composed of a wide spectrum of particles, which lose energy when transmitting through matter. Upon irradiation of a PSC, energy of an incident proton is lost through two mechanisms, elastic non-ionizing energy loss (NIEL) resulting in atomic displacements, and inelastic ionizing energy loss (IEL) which causes heating due to scattering with the surrounding electron cloud[5,10,11]. Protons are particularly important for radiation testing due to their high NIELs compared to electrons (Supplementary Fig. 1)[5,10,12]. Though alpha particles have higher NIELs, they have very low fluences in space orbits and are therefore not representative of the space environment. The ratio of IEL to NIEL (IEL/

[1]National Renewable Energy Laboratory (NREL), Golden, CO 80401, USA. [2]School of Chemistry and Materials Science, Rochester Institute of Technology, Rochester, NY 14623, USA. [3]Department of Physics, University of North Texas, Denton, TX 76203, USA. [4]Department of Applied Physical Sciences, University of North Carolina, Chapel Hill, NC 27599, USA. [5]NASA Glenn Research Center, Cleveland, OH 44135, USA. [6]Homer L. Dodge Department of Physics and Astronomy, University of Oklahoma, Norman, OK 73019, USA. [7]Department of Electrical Engineering, University at Buffalo, Buffalo, NY 14260, USA.
✉e-mail: ahmad.kirmani@rit.edu; joey.luther@nrel.gov

NIEL) increases with proton energy making higher energy protons (1 MeV and higher) less likely to create atomic displacements (Supplementary Fig. 2). The implications of IEL on PSC performance are not well understood, however, IEL has recently been suggested to cause local healing of the perovskite lattice damaged by NIEL as a result of low formation energies of perovskites coupled with stronger electron-phonon interactions compared to conventional semiconductors[5,13–15]. Radiation tolerance of conventional space PV is determined solely by atomic displacements due to NIEL[11], while IEL is not known to play any significant role. Developing this understanding for hybrid organic-inorganic semiconductors is critical to accurate determination of PSCs' radiation tolerance. Low-energy protons are critical for initial radiation testing of PSCs since they minimize IEL and the resulting heating and are the most prominent in space radiation environments[10]. The resulting passivation of defects associated with heating from high-energy protons could mask the true extent of radiation damage. Here, we directly demonstrate variance in performance from exposure to various proton energies.

In this work, we provide conclusive evidence of IEL-induced efficiency recovery in PSCs. We design a dual dose experiment where PSCs are first irradiated with low-energy protons (0.06 MeV) that result in degraded power-conversion-efficiency (PCE) due to NIEL-induced atomic displacements. Next, some PSCs are further irradiated with a high-energy proton beam (1.0 MeV) with a higher IEL/NIEL ratio. An increased PCE after this second irradiation confirms healing of the damaged perovskite lattices. Due to the strong electron-phonon interactions in perovskites and polaronic charge transport[14,16], IEL causes phonon vibrations in the lattice sufficient for driving the displaced atoms back to lattice positions, suppressing non-radiative recombination and increasing the PCEs. Further increase in the IEL component does result in a drop in PCE, thus highlighting the complicating factors of analyzing radiation-induced effects. Monte-Carlo ion-solid interaction simulations suggest that higher energy protons target different defect species, specifically the inorganic framework consisting of halogens, likely explaining the performance loss with increasing IEL. Note: while in some definitions "dose" could mean the amount of radiation absorbed, in this paper we use the terminology to simply imply a radiation exposure event[17]. This work solves an important puzzle regarding radiation-matter interactions in PSCs and further highlights that low-energy protons enable rapid screening of perovskites for long term performance in space.

## Results

PSCs featuring a 1.62 eV bandgap triple-cation perovskite active layer, $Cs_{0.05}(FA_{0.83}MA_{0.17})_{0.95}Pb(I_{0.83}Br_{0.17})_3$, were used in this study given their suitability as a standalone perovskite PV technology and relative stabilities[18]. Cells were fabricated on space-qualified quartz substrates. NIP device schematic is shown in Fig. 1A and highlights the various layers in the stack. The choice of proton energies and the dual dose irradiation conditions were informed by theoretically simulating the proton-PSC interaction using the Stopping and Range of Ions in Matter (SRIM) code[19]. SRIM is a binary collision Monte Carlo simulation that tracks the trajectory of an incident ion by quantifying the NIEL and IEL components as it traverses matter. Simulation details are provided in the Experimental Section. Device architecture and various layer thicknesses were defined in SRIM based on thickness information obtained from cross-sectional SEM (X-SEM). An image of a representative PSC is shown in Supplementary Fig. 3 and denotes the various layer thicknesses.

Figure 1B shows simulated interaction of two different proton energies with the PSC: 0.06 MeV and 1.0 MeV, modeled using SRIM. Each case shows the incidence and subsequent traversal of 100,000 protons (straggling) through the device. Overlaid on these straggling images are the total vacancies per proton created within the device for each proton energy scenario. 0.06 MeV protons create more vacancies

due to their high NIEL (Supplementary Fig. 4). Increasing the proton energy reduces the number of vacancies created and 1.0 MeV protons pass through the PSC with far less scattering.

The NIEL and IEL components of the proton energy obtained from SRIM are shown in Fig. 1C, D. Solid lines denote the cases of individual proton energies, while the dashed lines represent various scenarios of dual dose irradiation. To obtain these profiles, the fluence of the 1.0 MeV protons was first scaled to match the NIEL curve of 0.06 MeV protons at a standard relevant testing fluence. This was done by multiplying the 1.0 MeV curve by a factor of 37.7 indicating that 37.7 times higher fluence of 1.0 MeV protons would create the same NIEL as dosing with 0.06 MeV protons. Next, various linear fluence combinations of the 0.06 MeV curve and the scaled 1.0 MeV curve were plotted. The combinations are: 90% 0.06 MeV NIEL + 10% scaled 1.0 MeV NIEL (dashed black), 50% 0.06 MeV NIEL + 50% scaled 1.0 MeV NIEL (dashed blue), 10% 0.06 MeV NIEL + 90% scaled 1.0 MeV NIEL (dashed red). All these combinations deposited the same NIEL in the PSC, as Fig. 1C shows. The corresponding IEL curves for these combinations are shown in Fig. 1D. The IEL component is found to increase as the contribution of 1.0 MeV protons in the combination increases, while the NIEL remains constant as shown in Supplementary Fig. 5 where the cumulative sums of the NIEL and IEL within the device for each irradiation scenario are plotted. The IEL scaling factor (IEL x) shown in the legend denotes the relative increase in IEL compared to the control single dose irradiation scenario of 0.06 MeV protons. These dual dose scenarios allowed us to tune the IEL component over a wide range from 3.6- to 27.5-times the control, independent of the NIEL. Importantly, this platform can be used to explore an IEL-related phenomenon, such as the self-healing, without changing the vacancy profile in the PSCs incurred from NIEL.

We used these theoretical insights to inform the design of dual dose experiments and irradiated PSCs at normal incidence from the metal electrode side, in accordance with PSC radiation-testing guidelines published recently (Fig. 2)[10]. A NIEL-dominated single dose irradiation with 0.06 MeV protons at $1 \times 10^{13}$ cm$^{-2}$ fluence resulted in a reduction in $V_{OC}$ and FF. A PCE remaining factor (ratio of the PCE after and before irradiation) of 0.74 was obtained and is primarily the result of a reduced open-circuit voltage ($V_{OC}$) and fill-factor (FF). As expected, the NIEL-dominated protons create atomic displacements and lead to non-radiative recombination. Figure 3A shows representative J-V curves of some of the PSCs. Next, we irradiated a PSC using the dual dose scheme explained above. The device was first exposed to 0.06 MeV protons at a fluence of $0.9 \times 10^{13}$ cm$^{-2}$ and next with 1.0 MeV protons at a fluence of $3.8 \times 10^{13}$ cm$^{-2}$. When combined, this dual dose scheme delivers 3.6 times higher IEL and the same NIEL as the control NIEL-dominated single dose radiation. The PCE remaining factor increased to 0.83 with the FF showing an improvement. PSC irradiated with a dual dose scheme delivering further higher IELs resulted in performance degradation with $J_{SC}$ showing a decreasing trend. Remaining factors for $J_{SC}$, $V_{OC}$, FF, and PCE for various IEL scenarios are shown in Fig. 3B–E. The device parameters before and after irradiation are summarized in Table 1 and shown in Supplementary Fig. 6. Table 1 also summarizes the various single dose and dual dose irradiation conditions.

The largest increase in the PCE remaining factor was observed specifically when IEL $x$ = 3.6. As Fig. 3D demonstrates (black arrow), the FF remaining factor rises sharply at this point and approaches 0.9. This likely points toward suppression of non-radiative recombination in the PSC. Recent studies point toward IEL-induced electronic ionization to cause lattice vibrations and local heating in the perovskite lattice[5]. It has been argued that this local heating causes vacancies to be replenished, healing the lattice[5]. Displaced atoms in perovskite lattices can easily fill created vacancies due to low formation energies and high ionic mobility in these semiconductors. This is different from the case of conventional semiconductors (Si and III-V) where IEL does not play a

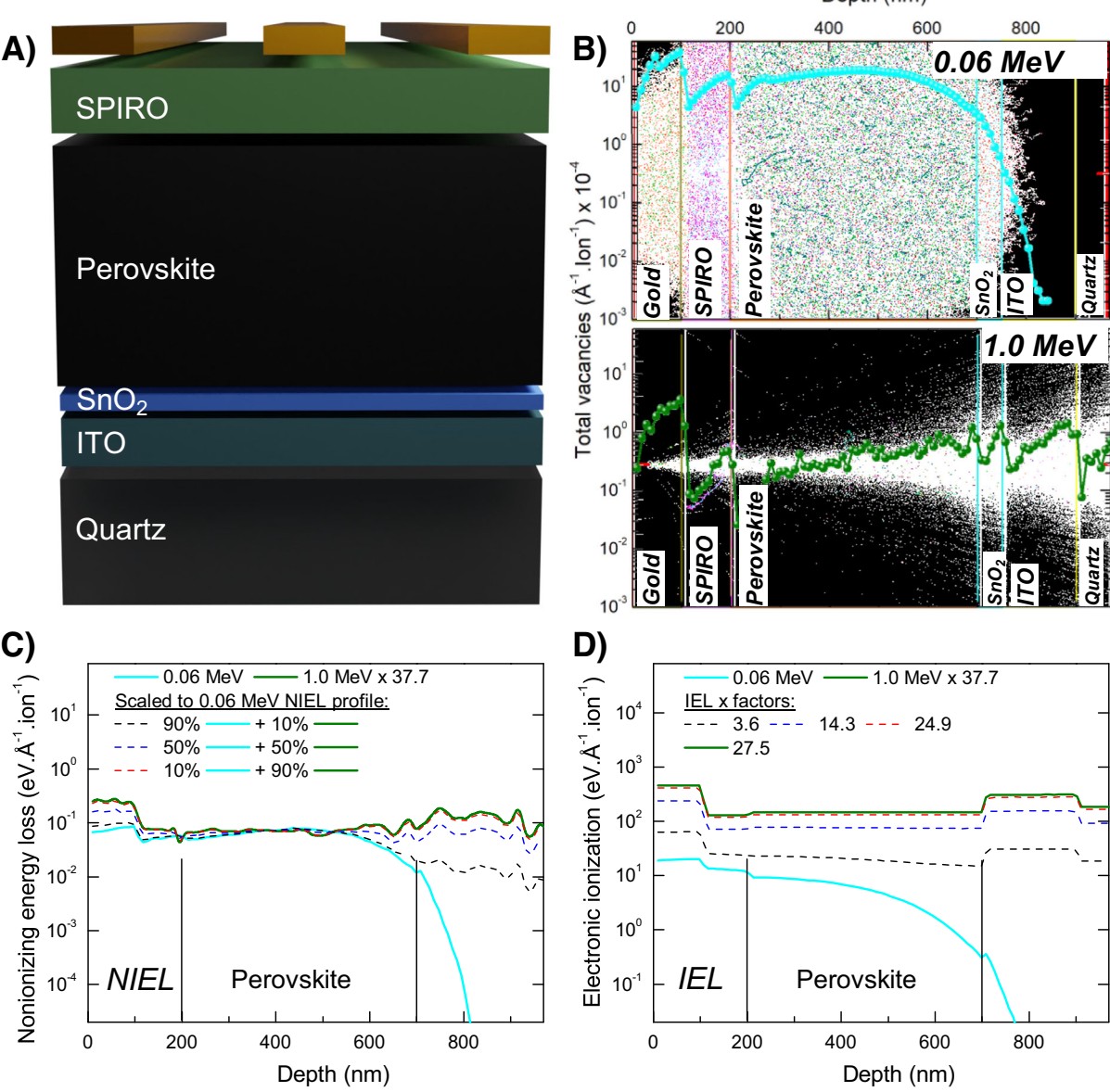

**Fig. 1 | SRIM simulations of proton-NIP PSC interactions. A** Device schematic showing various layers in the stack. **B** SRIM simulations showing proton straggling for 0.06 MeV (cyan) and 1.0 MeV (green) protons. Vacancy profiles are overlaid on the straggling images. NIEL and IEL profiles for the various irradiation scenarios considered are shown in (**C**) and (**D**). By selecting the two radiation conditions (energy and fluence) shown with dashed lines in (**C**), the NIEL created in each cell is the same, whereas the IEL increases up to a factor of 27.5 greater for increasingly higher energy dosing as shown in (**D**). This enables us to tune the IEL/NIEL ratio. Source data are provided as a Source Data file.

significant role in radiation-matter interactions and radiation tolerance of conventional space PV is primarily determined by NIEL[11].

We carried out thermal admittance spectroscopy (TAS) to probe the nature of defects caused by the irradiation and the healing process. TAS allowed us to extract the energetic distribution of charge carrier trap states in the perovskite bandgap. Figure 4A shows the trap density of states (tDOS) as a function of energy depth inside the bandgap for a non-irradiated control cell (black), a cell irradiation with only a 0.06 MeV proton beam (cyan), and a healed cell irradiated with dual dose radiation (blue). Three distinct trap bands centered around different energy depths can be observed in the spectra: band I centered around 0.26 eV, band II centered around 0.35 eV, and band III centered around 0.50 eV. In an earlier report, these trap bands have been shown to originate from defects in the perovskite absorber layer[20]. After irradiation, both trap bands I and III slightly decreased, while trap band II showed a 100% increase from $3 \times 10^{15}$ cm$^{-3}$ eV$^{-1}$ to $6 \times 10^{15}$ cm$^{-3}$ eV$^{-1}$.

This result indicates that the trap band II (deep traps) is more detrimental to PSC performance. Band II has been recently suggested to be caused by positively charged iodine interstitials in perovskites containing iodine[20]. For the healed device, the tDOS of all three bands recover and become similar to the non-irradiated control device, consistent with the PSC efficiency increase during the healing process. We note that in Fig. 4A the major changes observed are in trap bands I and II which are measured at AC frequencies in the range of 10 kHz–2 MHz. These are significantly higher than the response frequencies for drift and diffusion due to ionic defects (up to 1 kHz) which occur on timescales of milliseconds to seconds[21]. This implies that the defects measured here are different from ionic drift and diffusion.

Elemental vacancy profiles simulated using SRIM suggest that H and I atoms are the most displaced species by 0.06 MeV protons (Fig. 4B). Of these, the displaced H atoms are expected to be healed easily given their low masses compared to the heavier atoms of I. These

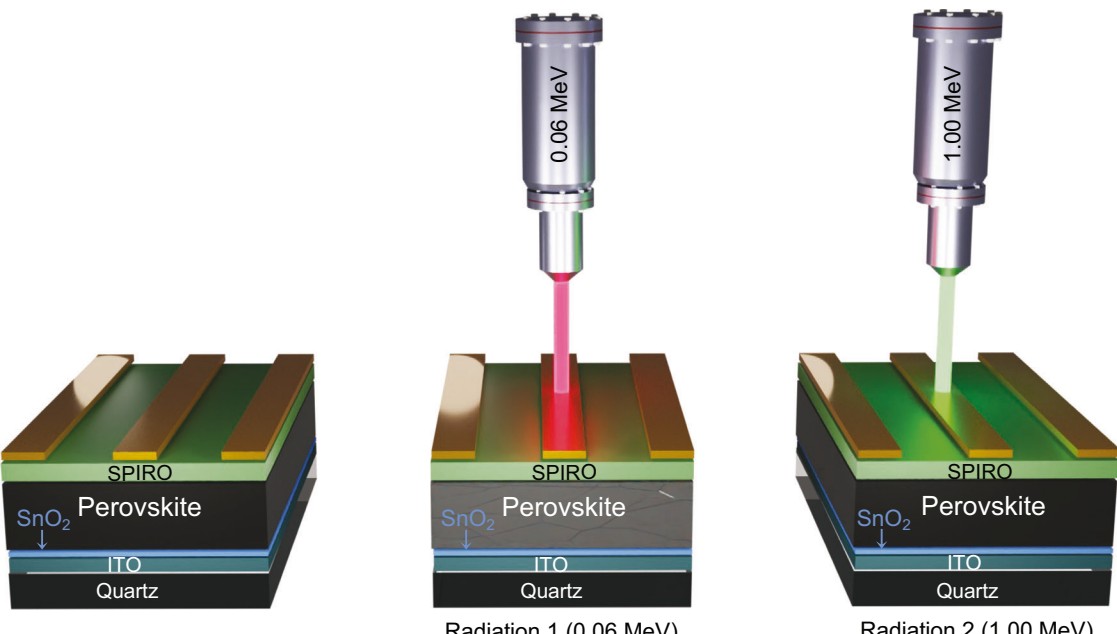

**Fig. 2 | Dual dose irradiation experiments.** Irradiation of the PSC with a NIEL-dominated 0.06 MeV proton beam (red) is followed by irradiation with a 1.0 MeV proton beam (green). By varying the fluence of the two radiation exposures, we selectively demonstrate how IEL participates in partial recovery of the solar cell performance after initial radiation damage.

H atoms are removed from the organic formamidinium and methylammonium cations at the A-site resulting in de-protonation of the molecules. The displaced I atoms expectedly result in I interstitials explaining the formation of trap band II as observed in the TAS measurements[20], and the resulting loss in PCE remaining factor to 0.74. For the dual dose irradiation scenarios, the extra IEL imparted to the perovskite lattice leads to phonon vibrations causing the displaced I atoms to move back to their initial lattice positions. This results in a drop in the band II tDOS from $6 \times 10^{16}$ cm$^{-3}$ eV$^{-1}$ to $3 \times 10^{16}$ cm$^{-3}$ eV$^{-1}$ and an increased PCE remaining factor of 0.83.

In organic semiconductors, electronic ionization has been shown to break C-H bonds and abstract H atoms resulting in defect creation[22]. Thermally activated H migration results in healing of these defects[22]. The processes of atomic displacement (NIEL) and electronic ionization (IEL) are correlated and together impact the irradiated lattice[23,24]. In a timespan of femto-picoseconds following irradiation, IEL raises the electronic temperature above that of the lattice causing localized inelastic thermal spikes pushing the system far from equilibrium[23]. As the system approaches thermal equilibrium over time, the electron transfers thermal energy to the lattice due to electron-phonon coupling, providing enough energy to the displaced atoms to reorganize. Impact of IEL on perovskite semiconductors can likely be explained via a similar mechanism given the presence of C-H and C-N species at the A-site.

SRIM simulations also provide insight into the performance loss observed for higher IEL scenarios. As Supplementary Fig. 7 shows, 1 MeV protons mostly targe the inorganic framework, predominantly creating I and Pb vacancies. Therefore, as the 1.0 MeV component is increased in the dual dose experiments to raise the IEL, the radiation inadvertently disrupts the inorganic framework and leads to vacancies that are relatively more challenging to heal[25]. This energy dependence of defect formation observed in our devices backed by theoretical modeling further highlights the complex nature of radiation-matter interactions in perovskite semiconductors that are strikingly different from conventional semiconductors.

Another possible explanation for the PCE loss observed for the higher IEL scenarios is the low tolerance of organic-containing perovskites to high temperatures[1]. Increased IEL can cause temperature increases via phonon vibrations above the threshold of these lattices resulting in permanent damage. Thus, while a slight IEL can heal damaged lattices, further increase can lead to excessive heating and further damage.

We note that SRIM simulations do not consider chemical bond strengths and the role bond strengths play in defining radiation effects remains to be understood. Therefore, while more work is needed to get a clearer picture of the defect creation and healing phenomenon in perovskites, the experiments we have designed induce and probe these processes and, together with theoretical modeling, present the best preliminary picture to date of the mechanisms involved.

To explore generality of the dual dose irradiation scheme, we also carried out these experiments for PSCs in the PIN architecture. SRIM simulations were again used to guide the choice of 0.06 MeV and 1.0 MeV proton radiation fluences (Supplementary Fig. 8). Cumulative sums of energy loss curves within the devices are shown in Supplementary Fig. 9 and confirm that while NIEL remains constant for these irradiation conditions, IEL scales with the 1.0 MeV component. J-V curves for a solar cell before irradiation, after irradiation with 0.06 MeV protons, and healed with dual dose irradiation are shown in Fig. 5A. Device schematic is shown in Fig. 5B, and an X-SEM image of a representative PIN PSC is shown in Supplementary Fig. 10. Remaining factors for the various device parameters for a series of IEL x scenarios are shown in Fig. 5C–F. Dual dose irradiation response of PIN PSCs is found to be similar to the case of NIP PSCs. While a single dose irradiation with 0.06 MeV protons degrades the PCEs leading to a remaining factor of 0.72, the dual dose irradiation condition corresponding to an IEL x of 2.4 results in remarkable healing of the devices leading to an enhanced PCE remaining factor of 0.85. Further increase in IEL expectedly leads to performance loss. These observations are summarized in Table 2.

Time-resolved photoluminescence (TRPL) measurements on these device stacks add support to the defect creation and healing mechanism described above. Data shown in Supplementary Fig. 11 and Supplementary Table 1 highlight a 50% reduction in the longer decay component ($\tau_2$) for the single dose irradiated stack (28.1 ns) compared to the

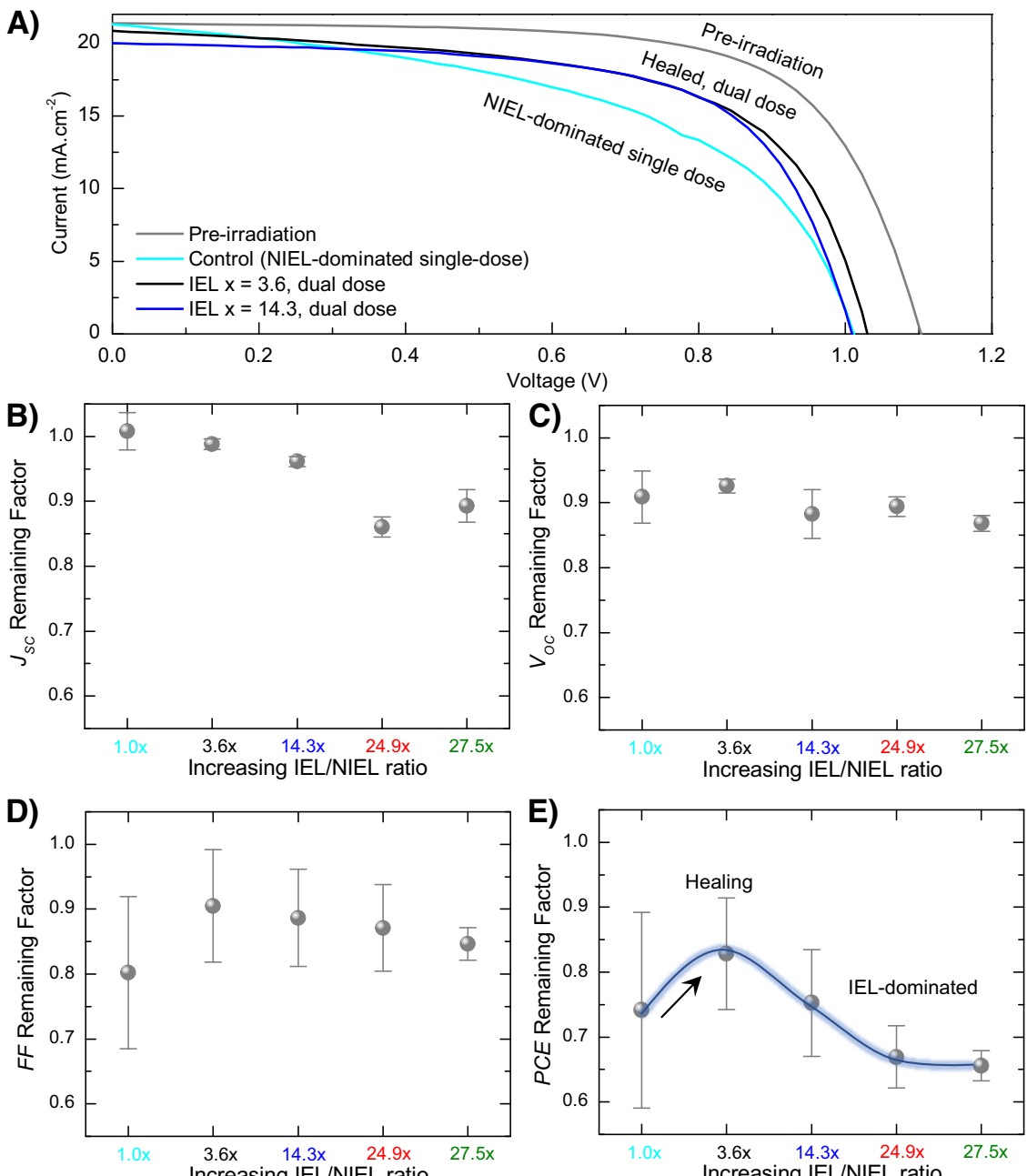

**Fig. 3 | Radiation response of proton-irradiated NIP PSCs. A** J-V curves of representative NIP solar cells prior to irradiation (gray), after irradiation with the NIEL-dominated single dose of 0.06 MeV protons (cyan), and after being healed with dual dose radiation (blue, black). Remaining factors for (**B**) $J_{SC}$, (**C**) $V_{OC}$, (**D**) FF, and (**E**) PCE for irradiated NIP PSCs for various IEL x scenarios. The single dose control device is denoted by the IEL x = 1.0 case. Black arrow indicates IEL-induced healing and subsequent increase in *PCE* for the IEL x = 3.6 case. All parameters are averages over 4–5 devices and error bars represent standard deviation. Source data are provided as a Source Data file.

non-irradiated sample (53.9 ns), indicating creation of non-radiative recombination centers. $\tau_2$ increases for the dual dose samples as IEL x is raised from 1.6 (36.8 ns) to 2.4 (46.8 ns) closer to the non-irradiated sample and pointing toward healing of the recombination centers. Taken alone, this may not prove the point because of the small differences in the lifetimes, but the trend is supportive of the hypothesis.

We were motivated to understand if the devices could still be healed using dual dose irradiation even when NIEL was not held constant. To test this, we considered three irradiation scenarios: control single dose irradiation 100% 0.06 MeV, 100% 0.06 MeV + 3% scaled 1.0 MeV, and 100% 0.06 MeV + 7% scaled 1.0 MeV. As can be seen, all the three *PIN* cells received the same dose of 0.06 MeV radiation. The

dual dose irradiation scenarios will expectedly have a higher NIEL since the 0.06 MeV fluence is held constant. Supplementary Fig. 12 shows the NIEL and IEL curves in the devices comparing the two dual dose irradiation schemes. NIEL curves in Supplementary Fig. 12C increase for the dual dose conditions. Supplementary Fig. 13 further highlights this point. PCE remaining factors of PIN solar cells exposed to these three irradiation conditions are shown in Supplementary Fig. 14. Remaining factor is found to slightly increase from 0.80 to 0.82 for the IEL x = 1.6 case suggesting that despite the increasing NIEL, the higher IEL associated with this dual dose condition can heal the damage. A further increase in IEL x to 2.4 increases the NIEL significantly such that the remaining factor decreases to 0.74.

**Table 1 | Summary of NIP device parameters for various irradiation conditions**

| Device | Rad. 1 (MeV, cm$^{-2}$) | Rad. 2 (MeV, cm$^{-2}$) | IEL x | $J_{SC}$ (mA cm$^{-2}$) | $V_{OC}$ (Volts) | FF | PCE (%) | PCE remaining factor |
|---|---|---|---|---|---|---|---|---|
| 1. | 0.06, 1×10$^{13}$ | – | – | 20.99 ± 0.46 | 1.11 ± 0.01 | 0.69 ± 0.04 | 16.00 ± 1.10 | 0.74 ± 0.15 |
| | | | | 21.24 ± 0.39 | 1.01 ± 0.05 | 0.56 ± 0.08 | 12.16 ± 2.41 | |
| 2. | 0.06, 0.9×10$^{13}$ | 1.0, 3.8×10$^{13}$ | 3.6 | 21.17 ± 0.16 | 1.11 ± 0.01 | 0.72 ± 0.02 | 16.91 ± 0.70 | 0.83 ± 0.09 |
| | | | | 20.92 ± 0.07 | 1.02 ± 0.01 | 0.65 ± 0.06 | 14.00 ± 1.34 | |
| 3. | 0.06, 0.5×10$^{13}$ | 1.0, 1.9×10$^{14}$ | 14.3 | 20.71 ± 0.10 | 1.11 ± 0.01 | 0.71 ± 0.02 | 16.29 ± 0.69 | 0.75 ± 0.08 |
| | | | | 19.91 ± 0.12 | 0.98 ± 0.04 | 0.63 ± 0.05 | 12.25 ± 1.23 | |
| 4. | 0.35, 1.1×10$^{14}$ | – | 16.0 | 21.33 ± 0.09 | 1.11 ± 0.01 | 0.69 ± 0.05 | 16.52 ± 1.33 | 0.67 ± 0.08 |
| | | | | 20.11 ± 0.45 | 0.98 ± 0.03 | 0.57 ± 0.05 | 11.17 ± 0.97 | |
| 5. | 0.06, 0.1×10$^{13}$ | 1.0, 3.4×10$^{14}$ | 24.9 | 21.14 ± 0.21 | 1.12 ± 0.01 | 0.73 ± 0.02 | 17.20 ± 0.72 | 0.67 ± 0.05 |
| | | | | 18.19 ± 0.27 | 1.00 ± 0.01 | 0.63 ± 0.04 | 11.51 ± 0.67 | |
| 6. | 1.0, 3.8×10$^{14}$ | – | 27.5 | 21.27 ± 0.13 | 1.12 ± 0.01 | 0.74 ± 0.02 | 17.63 ± 0.57 | 0.66 ± 0.02 |
| | | | | 18.99 ± 0.53 | 0.98 ± 0.01 | 0.62 ± 0.00 | 11.57 ± 0.16 | |

Parameters are averaged over 4–5 devices per condition. Error bars represent standard deviation. Top rows: pre-irradiation, Bottom rows: post-irradiation.

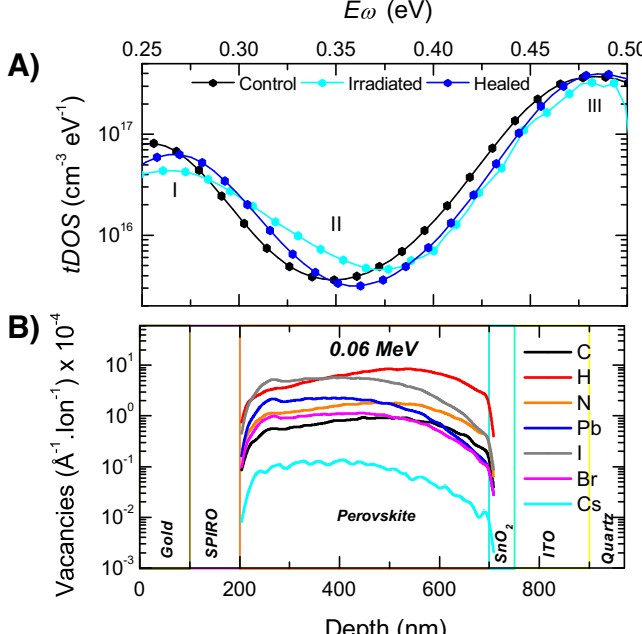

**Fig. 4 | Radiation-induced defect creation and healing. A** Trap density of states (tDOS) of the PSCs obtained using TAS prior to irradiation (black), after irradiation with the NIEL-dominated single dose of 0.06 MeV protons (cyan), and after being healed with dual dose radiation (blue) with IEL x = 3.6. **B** SRIM simulations showing elemental vacancy profiles within the perovskite active layer for 0.06 MeV protons. H (red) and I (gray) are the most displaced elements. Source data are provided as a Source Data file.

To further probe and elucidate the working mechanism of defect healing, we designed three independent tests: dark aging, AM 1.5 G light soaking, and thermal annealing. We posited that these conditions could provide the required thermal energy for rearrangement of displaced atoms. A non-irradiated, a single dose irradiated, and a dual dose irradiated device (IEL x = 1.6) were considered, and all the tests were carried out in a N$_2$ glove box. If any of these conditions were to promote healing, we would expect the single dose devices to recover in PCEs and approach the PCEs of the dual dose devices that have been healed by the 1.0 MeV proton radiation.

Supplementary Fig. 15 shows the effect of dark aging followed by AM 1.5 G light soaking. While the single dose device (as expected) started with a lower PCE remaining factor (0.59 ± 0.10) compared to

the dual dose device (0.65 ± 0.07), improvement was observed during dark aging. Over the course of 10 days in this condition, remaining factors for both devices increased to 0.77. This observation offers a key insight into the nature of radiation-induced defects in PSCs. These defects have low activation energies and undergo room temperature annealing over a sufficiently long time of 10 days. These devices were subsequently exposed to the simulated AM 1.5 G solar spectrum for a total of 15 min. Light soaking was not found to have any positive effect on the devices which showed a remaining factor drop due to light soaking and recovered to pre-light soaking values only after a day of dark aging.

Thermal conditioning was carried out on a separate set of PSCs to further understand the defect healing behavior observed for the dark aged devices. Given the tendency of these defects to heal at room temperature over multiple days, we expected this healing to accelerate and occur over a few hours at elevated temperatures. Thermal annealing data is displayed in Supplementary Fig. 16 which shows $J_{SC}$, $V_{OC}$, FF, and PCE remaining factors as a function of annealing time for 60 °C and 90 °C. Annealing was carried out in a N$_2$ glove box and the devices were tested after 2, 5, and 12 h. While the PCE remaining factor of the dual dose device largely remained unchanged after 12 h of 60 °C annealing, the single dose device showed an obvious improvement[6]. In fact, this device recovered and achieved a similar remaining factor (~0.85) as the dual dose device after 5 h. This is clear evidence of thermally activated healing of radiation-induced defects and substantiates the observation that the IEL-dominated 1.0 MeV protons affect phonon-mediated repositioning of the atoms initially displaced by the NIEL-dominated 0.06 MeV protons leading to damage recovery.

The 90 °C annealing experiments offer another key insight into thermal behavior of irradiated PSCs. The single dose devices are found to undergo significant degradation within 2 h of annealing at this temperature. Surprisingly, the dual dose devices show a markedly higher thermal resilience. While only a few minutes of annealing at this high temperature might be enough for healing, it appears that the initial defect density also dictates temperature stability. The reduced defect density in the dual dose devices results in a higher temperature resilience compared to the single dose devices. Suppressing radiation-induced defects can therefore prolong the lifetime of PSCs in space environments where harsh temperatures are expected. The non-irradiated devices with the least defect density do not show any performance loss after 12 h of annealing.

While this study provides direct evidence of defect creation and healing in the perovskite active layer, the possibility that the other device layers and interfaces might also be contributing to the observed radiation response cannot be ruled out[26]. For example, the

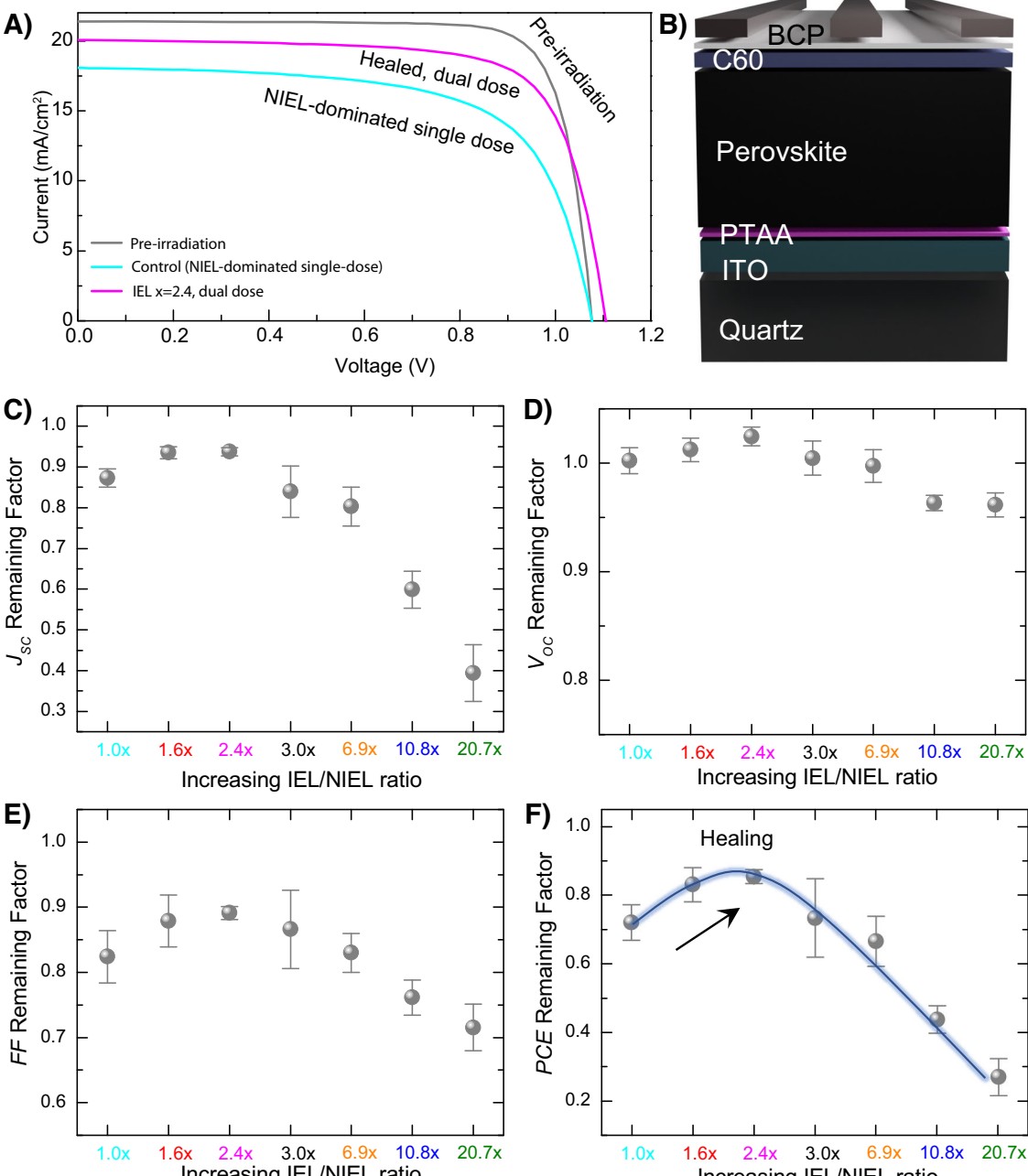

**Fig. 5 | Radiation response of proton-irradiated PIN PSCs. A** J-V curves of representative PIN solar cells prior to irradiation (gray), after irradiation with the NIEL-dominated single dose of 0.06 MeV protons (cyan), and after being healed with dual dose radiation (magenta). **B** Device schematic. Remaining factors for (**C**) $J_{SC}$, (**D**) $V_{OC}$, (**E**) FF, and (**F**) PCE for irradiated PIN PSCs for various IEL x scenarios.

The single dose control device is denoted by the IEL $x = 1.0$ case. Black arrow indicates IEL-induced healing and subsequent increase in PCE for the IEL $x = 2.4$ case. All parameters are averages over 4–5 devices and error bars represent standard deviation. Source data are provided as a Source Data file.

bis(trifluoromethane)sulfonimide lithium (Li-TFSI) salt used to dope the spiro-OMeTAD hole transporter for NIP PSCs is known to actively migrate across the device stack[27,28]. It is possible that proton irradiation displaces Li-TFSI resulting in de-doping of spiro-OMeTAD. It is worth noting that in addition to the initial reduction and subsequent healing observed for $V_{OC}$, a similar trend was observed for FF. A loss in FF is linked to non-radiative recombination losses in the active layer and charge transport losses at the carrier extraction interfaces[29]. Therefore, while recombination in the perovskite layer can reduce carrier diffusion length limiting FF and $J_{SC}$, radiation-induced increase in resistance of transport layers can also explain the observed FF changes[30]. Importantly, for ideality factors (n) exceeding

1, non-radiative recombination causes a drop in both the $V_{OC}$ and FF[31]. It is therefore expected that by healing non-radiative recombination and increasing the $V_{OC}$, the IEL-dominated 1 MeV protons also improve FF.

Future studies aiming to specifically probe the interface effects will require targeted proton irradiation by accordingly tuning the incident proton energy. Supplementary Fig. 17 shows vacancy profiles in PSCs from incident protons beams with energies of 0.01, 0.03, and 0.06 MeV. In total, 0.01 MeV protons mostly damage the hole-transporting interface, while the 0.03 MeV protons create vacancies mostly in the perovskite active layer without affecting the electron transporting interface. These targeted irradiation experiments can be used

**Table 2 | Summary of PIN device parameters for various irradiation conditions**

| Device | Rad. 1 (MeV, cm⁻²) | Rad. 2 (MeV, cm⁻²) | IEL x | $J_{SC}$ (mA cm⁻²) | $V_{OC}$ (Volts) | FF | PCE (%) | PCE remaining factor |
|---|---|---|---|---|---|---|---|---|
| 1. | 0.06, $1 \times 10^{13}$ | – | – | 21.39 ± 0.15 | 1.07 ± 0.01 | 0.81 ± 0.00 | 18.72 ± 0.27 | 0.72 ± 0.05 |
| | | | | 18.68 ± 0.18 | 1.08 ± 0.00 | 0.67 ± 0.03 | 13.51 ± 0.96 | |
| 2. | 0.06, $0.97 \times 10^{13}$ | 1.0, $1.13 \times 10^{13}$ | 1.6 | 21.31 ± 0.16 | 1.08 ± 0.01 | 0.81 ± 0.01 | 18.66 ± 0.27 | 0.83 ± 0.05 |
| | | | | 19.93 ± 0.28 | 1.09 ± 0.00 | 0.71 ± 0.03 | 15.53 ± 0.91 | |
| 3. | 0.06, $0.93 \times 10^{13}$ | 1.0, $2.64 \times 10^{13}$ | 2.4 | 21.33 ± 0.18 | 1.07 ± 0.01 | 0.81 ± 0.01 | 18.50 ± 0.28 | 0.85 ± 0.02 |
| | | | | 19.99 ± 0.12 | 1.10 ± 0.01 | 0.72 ± 0.01 | 15.83 ± 0.28 | |
| 4. | 0.06, $0.90 \times 10^{13}$ | 1.0, $3.8 \times 10^{13}$ | 3.0 | 21.31 ± 0.14 | 1.07 ± 0.00 | 0.80 ± 0.01 | 18.46 ± 0.46 | 0.73 ± 0.11 |
| | | | | 17.89 ± 1.33 | 1.08 ± 0.02 | 0.70 ± 0.04 | 13.57 ± 2.07 | |
| 5. | 0.06, $0.70 \times 10^{13}$ | 1.0, $1.13 \times 10^{14}$ | 6.9 | 21.31 ± 0.10 | 1.07 ± 0.01 | 0.81 ± 0.01 | 18.35 ± 0.15 | 0.67 ± 0.07 |
| | | | | 17.11 ± 1.02 | 1.06 ± 0.01 | 0.67 ± 0.02 | 12.24 ± 1.34 | |
| 6. | 0.06, $0.50 \times 10^{13}$ | 1.0, $1.9 \times 10^{14}$ | 10.8 | 21.27 ± 0.09 | 1.08 ± 0.01 | 0.81 ± 0.00 | 18.50 ± 0.17 | 0.44 ± 0.04 |
| | | | | 12.73 ± 0.98 | 1.04 ± 0.00 | 0.61 ± 0.02 | 8.12 ± 0.74 | |
| 7. | 1.0, $3.8 \times 10^{14}$ | – | 20.7 | 20.86 ± 0.31 | 1.07 ± 0.01 | 0.79 ± 0.02 | 17.54 ± 0.71 | 0.27 ± 0.05 |
| | | | | 8.22 ± 1.46 | 1.03 ± 0.00 | 0.56 ± 0.02 | 4.77 ± 0.93 | |

Parameters are averaged over 4–5 devices per condition. Error bars represent standard deviation. Top rows: pre-irradiation, Bottom rows: post-irradiation.

to decouple the irradiation effects on charge transport layers and the active layer.

It is clear from the above experiments that the device architectures where NIEL creates significant damage can be locally healed via a dual dose irradiation scheme. As such, devices that are less sensitive to irradiation may not have as pronounced of an effect. We carried out dual dose experiments on device architectures which involved modified charge transport layers, passivation, and packaging, and we have found these structures to be less radiation sensitive. Here IEL was found to have a less pronounced healing effect given the lower NIEL-induced defect density. These findings are summarized in Supplementary Figs. 18–20, Supplementary Table 2 and Supplementary Notes 1–2.

In summary, we provide the first direct proof of radiation-induced efficiency recovery in perovskite solar cells. The design of dual dose proton irradiation experiments allows us to uniquely tune radiation-matter interactions in the devices. We increase the ionization energy loss (IEL) of incident protons in a controlled manner and find direct evidence of healing via electronic ionization. Defects created by the damaging low-energy protons (0.06 MeV) are partially healed when the device is irradiated by an optimal dose of higher-energy protons (1.0 MeV). This causes an increase in the PCE remaining factor from 0.74 to 0.83 for NIP devices and 0.73 to 0.85 for PIN devices. These experiments reinforce the growing understanding that radiation-matter interactions in perovskite solar cells are very different from conventional space PV and present dual dose irradiation experiments as a unique platform to tune and heal defect densities within perovskite thin-films.

## Methods
### SRIM simulations
SRIM simulations were performed considering 100,000 protons using the 'full damage cascade' calculation mode. Following device structure was considered:

Au (100 nm)/spiro-OMeTAD (200 nm)/$Cs_{0.05}(MA_{0.17}FA_{0.83})_{0.95}Pb(I_{0.83}Br_{0.17})_3$ (500 nm)/$SnO_2$ (50 nm)/ITO (150 nm)/Glass (70 nm).

Proton were irradiated from the gold electrode side. A low glass substrate thickness was considered to speed up calculations.

Mass densities used were: $SiO_x$ = 2.13 g cm⁻³, $Al_2O_3$ = 3.95 g cm⁻³, $ZrO_2$ = 5.68 g cm⁻³, $HfO_2$ = 9.68 g cm⁻³, Au = 19.31 g cm⁻³, spiro-OMeTAD = 1.40 g cm⁻³, PTAA = 1.40 g cm⁻³, C60 = 1.65 g cm⁻³, BCP = 1.2 g cm⁻³, $Cs_{0.05}(MA_{0.17}FA_{0.83})_{0.95}Pb(I_{0.83}Br_{0.17})_3$ = 4.30 g cm⁻³, $SnO_2$ = 6.95 g cm⁻³, ITO = 7.20 g cm⁻³, Glass = 2.53 g cm⁻³.

Displacement energies used were: Au = 25 eV, C = 28 eV, H = 10 eV, N = 28 eV, O = 28 eV, Cs = 25 eV, Pb = 25 eV, I = 25 eV, Br = 25 eV, Si = 15 eV, In = 25 eV, Sn = 25 eV, Al = 25 eV, Zr = 25 eV, Hf = 25 eV.

### Materials
Lead (II) iodide (PbI₂; 99.99%) and lead (II) bromide (PbBr₂; >98.0%) were purchased from TCI America. Formamidinium iodide (FAI;) and methylammonium bromide (MABr;) were purchased from GreatCell Solar Materials. Bis(trifluoromethane)sulfonimide lithium salt (Li-TFSI) was purchased from Alfa Aesar. Cesium iodide (CsI; 99.999%) and 4-tert-butylpyridine (tBP; 96%) was purchased from Sigma-Aldrich. 2,2′,7,7′-tetrakis(N,N-di-p-methoxyphenylamino)-9,9′-spirobifluorene (spiro-OMeTAD; ≥99.5%) was purchased from Lumtec, $SnO_2$ colloidal dispersion was obtained from Alfa Aesar and diluted to 1.5% in DI water before use. Poly(triaryl amine) (PTAA; 5–20 kDa) was obtained from Solaris Chem, and poly(9,9-bis(3′-(N,N-dimethyl)-N-ethylammoinium-propyl-2,7-fluorene)-alt-2,7-(9,9-dioctylfluorene))dibromide (PFN-Br) was purchased from 1-Material. All the solvents (dimethylformamide (DMF), dimethylsulfoxide (DMSO), chlorobenzene (CB), toluene, methanol) were obtained from Sigma-Aldrich.

### Triple-cation perovskite active layer fabrication
PbI₂, PbBr₂, MABr, FAI, and CsI precursors were mixed in 1 ml DMF:DMSO solvent mixture (4:1 v:v) and vortexed to form a 1.26 M ink. The ink was filtered using a 0.45 mm nylon filter. In total, 50 μl ink dropped on the substrate was spun at 1000 r.p.m for 10 s followed by 6000 r.p.m for 20 s. In total, 150 μl CB was dropped in a continuous stream at the spinning substrates with 5 s remaining in the end of the spin cycle. This antisolvent rinse step changed the appearance of the spinning film from transparent to mild orange. After completion of the spin cycle, the substrate was immediately placed at a hot plate set at 100 °C for 60 min.

### Device fabrication
Quartz substrates (Ted Pella) with dimensions of 25.4 mm × 25.4 mm × 1 mm with ITO deposited in-house at NREL (sheet resistance ~15 Ω/□) were cleaned by sequential sonication in acetone (15 min) and isopropanol (15 min) purchased from Sigma-Aldrich. Substrates were blow-dried with nitrogen followed by 10 min of UV-ozone.

For NIP solar cells, deposition of the $SnO_2$ electron transporter was carried out in ambient. In total, 150 μl $SnO_2$ colloid was dropped on each substrate followed by spin-coating at 3000 r.p.m for 15 s. The coated substrates were placed at a hotplate set at 150 °C for 30 min.

This was followed by a further 10 min UV-ozone after which the substrates were transferred to a $N_2$ glove box where perovskite active layer fabrication was completed. spiro-OMeTAD was next deposited by dynamically spinning 10 μl of spiro-OMeTAD solution at 5000 r.p.m for 15 s. This solution was made right before deposition by dissolving 36.1 mg spiro-OMeTAD, 14.4 μl tBP and 8.8 μl Li-TFSI (520 mg in 1 ml acetonitrile) in 0.5 ml CB.

For PIN solar cells, deposition of the PTAA hole transporter was done after transferring the UV-ozone cleaned substrates to a $N_2$ glove box. In total, 50 μl of PTAA solution (1.5 mg ml$^{-1}$ in toluene) was spin-coated at 5000 r.p.m for 30 s followed by thermal annealing for 10 min at 100 °C. In total, 80 μl PFN-Br solution (0.5 mg ml$^{-1}$ in methanol) was then dynamically spun on these substrates at 5000 r.p.m for 30 s. This was followed by deposition of the perovskite active layer, as detailed above.

## Thermal evaporation
For NIP devices, 100 nm gold was evaporated at 0.5 Å s$^{-1}$ for the first 10 nm and 2.0 Å s$^{-1}$ for the remaining 90 nm. For PIN devices, 25 nm $C_{60}$ (0.30 Å s$^{-1}$), 6 nm BCP (0.15 Å s$^{-1}$) and 100 nm silver (0.5 Å s$^{-1}$ for the first 10 nm and 2.0 Å s$^{-1}$ for the remaining 90 nm) were evaporated.

## J-V characterization
PSCs were measured at room temperature in a $N_2$ glove box with a source meter (Keithley 2420) using a solar simulator (Newport, Oriel Class AAA, 94063 A) at 100 mW cm$^{-2}$ illumination (AM 1.5 G). The simulator was calibrated using an NREL-calibrated Si photodiode and a KG2 filter. Devices were measured in reverse scan (1.4 V to −0.2 V) and forward scan (−0.2 V to 1.4 V) at a scan speed of 0.464 V s$^{-1}$ and step size of 0.02 V. Device had an active area of 0.1 cm$^2$ and metal aperture (0.058 cm$^2$) during light illumination and J-V characterization.

## Thermal admittance spectroscopy (TAS)
TAS was performed using an Agilent E4980A precision LCR meter. During the measurement, the DC bias (V) was fixed at 0 V and the amplitude of the AC bias (δV) was 20 mV. The scanning range of the AC frequency ($f$) was 0.02–2000 kHz. The tDOS ($N_T$ ($E_\omega$)) was calculated using the equation $N_T$ ($E_\omega$) = (−1/$qkT$) ($\omega dC/d\omega$)•($V_{bi}/W$), where $q$, $k$, $T$, $\omega$ and $C$ are elementary charge, Boltzmann's constant, temperature, angular frequency, and specific capacitance, respectively. $W$ and $V_{bi}$ are the depletion width and build-in potential, respectively. The demarcation energy $E_\omega = kTln(\omega_O/\omega)$ (where $\omega_0$ is the attempt-to-escape angular frequency and equals $2\pi v_0$ $T^2$) is derived from the temperature-dependent C–f measurements for a perovskite with a similar composition reported in a previous work where $\omega_0 = 1 \times 10^{11}$ Hz[32]. Note the $\omega_0$ only determines the energy depth of the trap states from the band edges, and not the impact of relative trap density changes of the device upon irradiation and healing processes.

## SEM characterization
SEM imaging was performed on a Hitachi 4800 Field Emission Electron Microscope. Samples were mounted with carbon paint, and imaging executed at working distance range of 5–8 mm, as recommended by the vendor. Due to the volatility of perovskites, conservative imaging parameters were utilized, with lower energy and lower current settings. In cross-sectional orientation, a thin layer of gold was applied to mitigate charging effects.

## TRPL measurements
Samples were excited with an NKT supercontinuum laser (SuperK EXU-6-PP) operating at 500 nm. Emission spectra were recorded over a time period of 100 ns using a Hamamatsu C-10910-04 streak camera. The fluence of excitation was kept at ~1 μW/mm$^2$ with a repetition rate of 3.5 MHz.

## Proton irradiation
Proton irradiation was performed at the University of North Texas (UNT) Ion Beam Laboratory (IBL). The 0.06 MeV proton beams were extracted from a TiH solid cathode with a Source of Negative Ions by Cesium Sputtering (SNICS-II, NEC) associated with a 3 MV tandem accelerator (NEC 9SDH-2)[33,34]. The momentum analyzed proton beams were electrostatically raster scanned over the samples for uniform irradiation in a low-energy irradiation facility before injecting into the tandem accelerator. The proton flux was kept to similar levels that did not result in spatial variation or substantial heating of different target materials. In total, 0.06 MeV protons were irradiated with a beam flux of 18 nA/(3 cm$^2$ s) = 3.75 × 10$^{10}$ p/cm$^2$ s. In total, 0.35 MeV and 1.0 MeV protons extracted from a single ended accelerator (NEC 9SH) were irradiated with a beam flux of 60 nA/(4 cm$^2$ s) = 9.375 × 10$^{10}$ p/cm$^2$ s. All ion irradiations occurred under a vacuum of 1 × 10$^{-7}$ torr.

## Reporting summary
Further information on research design is available in the Nature Portfolio Reporting Summary linked to this article.

## Data availability
All of the data generated or analyzed during this study are included in the published article and its Supplementary Information files. Source data are provided with this paper.

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

## Acknowledgements

This work was authored in part by the National Renewable Energy Laboratory, operated by Alliance for Sustainable Energy, LLC, for the U.S. Department of Energy (DOE) under Contract No. DE-AC36-08GO28308. The device fabrication work was supported by the Operational Energy Capability Improvement Fund (OECIF) of the U.S. Department of Defense (DOD). The radiation dosing was supported in part by National Science Foundation (NSF) grant number HBCU-EiR –2101181 and ECCS-2210722. Defect density measurements were supported from the Center for Hybrid Organic Inorganic Semiconductors for Energy (CHOISE), an Energy Frontier Research Center funded by the Office of Basic Energy Sciences, Office of Science, within the US Department of Energy. K.V. and L.M.B. are grateful to the NASA Space Technology Mission Directorate for supporting this work through the 2019 Early Career Initiative. We thank Nancy Haegel for helpful discussions. The views expressed in the article do not necessarily represent the views of the DOE or the U.S. Government.

## Author contributions

A.R.K. and J.M.L. conceived and supervised this work and wrote the manuscript, with inputs from I.R.S. and B.R. A.R.K. fabricated the solar cells, carried out SRIM simulations, determined the relevant dual dose irradiation parameters, and performed device characterization. Proton irradiation was done by T.A.B., D.K.S. and B.R. at the University of North Texas. Z.N. and J.H. performed thermal admittance spectroscopy at the University of North Carolina. K.V. and R.S. assisted with thermal conditioning tests and TRPL, respectively. K.V. was supervised by L.M.B. All authors contributed to discussion and editing of the paper. X.Z. provided devices for Supplementary Fig. 18, and T.B.K. assisted with explanation for the FF improvement with IEL.

## Competing interests

The authors declare no competing interests.
