## [Peer Review File · Nature Communications]

Unraveling radiation damage and healing mechanisms in halide perovskites using energy-tuned dual irradiation dosingREVIEWER COMMENTS

Reviewer #1 (Remarks to the Author):

The manuscript by Kirmani et al. investigates the different damage mechanism of high-energetic irradiation on the performance of perovskite solar cells. This is directly important for future space applications, and, beyond that provides an insight into healing mechanisms in halide perovskites that could be important for other aspects/fields as well. The manuscript is well written, and the experiment dedicated to isolate potential defect healing due to local heating, is very well designed. In my opinion the paper will be highly interesting for the scientific community, and I'd be happy to recommend an improved version of the paper for publication.

Open questions:

- In the manuscript you focused on healing induced by the second irradiation with higher IEL/NIEL fraction. And while the experimental design with keeping the NIEL damage constant is extremely elegant I was wondering if you observe healing, e.g. upon thermal annealing, light illumination or gamma irradiation as well ?

- The mechanism that "IEL causes phonon vibrations in the lattice sufficient for driving the displaced atoms back to their lattice position" seems plausible, however I was wondering if other explanations exist/can be ruled out: For example one could (hypothetically) argue that the higher proton irradiation interacts more with the inorganic lattice, thereby creating defects that are different in nature. Then in a second step, created defects could interact.... potentially generating defect clusters /phases that (again hypothetically) are more benign than the individual defects. Moreover, creating of PbI₂ can act as a passivation while at higher perovskite/PbI₂ ratios device performance can degrade. See e.g. Q. Chen, H. Zhou, T.-B. Song, S. Luo, Z. Hong, H.-S. Duan, L. Dou, Y. Liu, Y. Yang, *Nano Lett.* 2014, 14, 4158. Or L. Chen, J. Chen, C. Wang, H. Ren, Y. Luo, K. Shen, Y. Li, F. Song, X. Gao, J. Tang, 2021, DOI 10.1021/acscami.1c02929. or T. J. Jacobsson, J.-P. Correa-Baena, E. Halvani Anaraki, B. Philippe, S. D. Stranks, M. E. F. Bouduban, W. Tress, K. Schenk, J. Teuscher, J.-E. Moser, H. Rensmo, A. Hagfeldt, *J. Am. Chem. Soc.* 2016, 138, 10331.

- Especially in the nip structure the main effect in PCE variation (degradation and healing) is from the FF. Why is this the case ? Trap assisted recombination should increase non-radiative recombination, and hence one could expect also a decrease of the Voc, maybe in conjunction with a decrease in luminescence (be it photo or electro-luminescence) ... Do you observe increased non-radiative recombination/Can you exclude other transport/charge extraction issues as main cause for the drop in FF/JSC?

- Could degradation of the spiro-OMeTAD also play a role? Especially since the impact on FF in case of the nip device is quite strong? See e.g. J. Barbé, D. Hughes, Z. Wei, A. Pockett, H. K. H. Lee, K. C. Heasman, M. J. Carnie, T. M. Watson, W. C. Tsoi, Sol. RRL 2019, 1900219, 1.

- Are there any conclusions that can be drawn from these investigations to direct future irradiation studies ?

- Lastly I have a question regarding the defect spectroscopy: Just recently Futscher et al (<https://pubs.acs.org/doi/full/10.1021/acsenergylett.1c02076>) showed that “ In mixed ionic–electronic conductors such as HaPs or CISs, the ionic responses often mask electronic defects, regardless of whether a voltage pulse, a current pulse, or a light pulse causes the disturbance, or whether current, capacitance, or voltage is the quantity to be observed.” and then concluded that one has to be careful that measured defect bands are due to electronic defects (excluding ionic) as otherwise erroneous conclusions may be drawn.

Reviewer #2 (Remarks to the Author):

This work by A. Kirmani et al. reports the degradation and healing behaviors of lead halide perovskite solar cells under energy-tuned proton radiation sources. A low-energy of 0.06 MeV and a high-energy of 1 MeV protons were used in this study. The radiation-matter interactions, energy loss, and vacancy densities of the exposed perovskite films were studied using SRIM simulations. The device performances and defect densities were also examined experimentally. The main discovery is the radiation-induced efficiency recovery under high-energy radiations with optimal doses, which is likely due to electronic ionization, lattice vibrations and local heating. This work provides important insights on the radiation effects on perovskite solar cells for space applications. I can recommend publication after the following comments and concerns being addressed.

1. Based on Fig S1, it seems that the difference in IEL/NIEL for 0.06 and 1 MeV is not very large. What about even lower or higher proton energies? For example, 0.01 eV vs 10 MeV?
2. Since the devices will be exposed to radiation only when they are used in space, I'm wondering if the experiments were carried out in a simulated space environment?
3. In page 6, the authors claimed “the NIEL-dominated protons create atomic displacements and leads to non-radiative recombination”. But there is no any direct proof of increase or decrease in non-radiative recombination. Can PL and TRPL be measured?

4. Because IEL may cause lattice vibration and local heating, can the local temperature be measured and quantified?
5. I noticed the initial device efficiencies are quite low (16~18%). Why not using higher performance devices in this study? – many of the NREL experts can make >24% PCE.
6. A little more background introduction will be helpful. For example, why proton radiation is selected in this study? Compared with other radiations, is proton a major concern in space? What happens when the devices are exposed to a combination of high, mid, low protons, together with X-ray, gamma-ray, and alpha and beta rays?

Reviewer #3 (Remarks to the Author):

Kirmani et al show results of perovskite solar cells that have been exposed to different proton irradiation energies and dosages. Notably the study compares two of the most common device architectures in a systematic study. The authors mainly use JV characteristics to determine whether that has been a healing effect. The study however lacks any depth in terms of mechanistic insights in the physical and chemical processes that occur in the devices before and after radiation. There is also a lack of materials characterization of the perovskite material itself. A recent study by Durant et al (<https://pubs.acs.org/doi/full/10.1021/acsenergylett.1c00756>) is a similar study in terms of tuning the proton energy and fluence and comparison of JV characteristics. Moreover, there is no indication on what step researchers need to take to manipulate and enhance the self healing mechanism to fully reproducibly recover the power conversion efficiency. Because of the lack of insight into any degradation and healing processes that are occurring in the perovskite absorber itself, this manuscript as it is, does not meet the requirements, in terms of broad appeal and of significance to the field, to be published in Nature Communications. Below are more detailed comments and questions regarding the data and its interpretation.

1. The level of detail of processes that are induced by the inelastic ionizing energy loss (IEL) mechanism are lacking. Considering the manuscript and the results are predominately attributed to this mechanism a much in-depth literature review needs to be carried out.
2. Following on from the previous point, if indeed the healing mechanism is associated with the warming of the lattice through phonons, then the authors must show the effect of annealing the device at a set temperature before and after proton irradiation can also help the healing process. Heat can also be dissipated into the lattice from hot carriers. To validate that it is indeed phonon mediated heating, the authors could excite the films with blue light after proton irradiation. The carrier cooling mechanisms, via phonons can also increase the lattice temperature.

3. What is the actual relationship between the fluence and NIELs and IELs? There is no clear evidence that increasing the fluence by a linear amount would result in the same proportionate increase in NIELs over IELs. This assumption seems arbitrary, without any scientific basis.

4. On page 6 the authors mention 'Importantly, this platform can be used to explore an IEL-related phenomenon, such as the self-healing, without changing the vacancy profile in the PSCs incurred from NIEL.' What is the vacancy profile induced by NIELs? Please may the author back this statement with a literature review.

5. The authors have recently published a paper on SiO₂ encapsulation for radiation protection of perovskite devices (<https://www.nature.com/articles/s41560-022-01189-1>). Therefore, there should be experimental evidence and comments on which proton energy beam would be required to replicate the damage of the radiation in the perovskite devices with encapsulation. This is especially important because a perovskite device will not be employed into space without encapsulation, therefore the trajectory and proton energy will be different with and without encapsulation.

6. Whilst the study is based on perovskite devices, there is very little characterization of the perovskite thin-film itself. The manuscript extensively talks about self-healing of the perovskite thin-film and attribute IEL to just the perovskite material, however the device is made up of many materials and interfaces, and therefore the authors need to investigate to find out exactly which layer and or interface is being damaged and or 'self-healed'.

7. The authors state 'Expectedly, the NIEL-dominated protons create atomic displacements and leads to non-radiative recombination.' There is no experimental evidence in the manuscript that show there is an increase in non-radiative recombination. Please may the authors provide data to back this assumption up. For example, time-resolved photoluminescence traces.

8. Please state how many devices were included for all the data and for the error analysis. For example, in Supplementary Figure 5.

9. For the NIP device the observed trend is not statistically significant. Why is this? Majority of the changes that occur in the device are from the Fill Factor. Why is this assigned to the suppression of non-radiative recombination? Usually, a change in recombination is mostly exhibited in a voltage loss. What about changes at the interface?

10. Please may the authors expand and give a description of what the potential mechanism is for the healing. From what is implied it seems that an atom is displaced from a NIELs event, followed by a lattice heating allowing that atom to move back. If this is the case then is the assumption that only one type of

atom is displaced? If so then is IEL providing sufficient energy for only these 'types' of atoms to move back into the vacancy?

11. From the tDAS data and the SRIM analysis the authors infer that H and I atoms are expected to be displaced most. The following explanation on the healing has very little chemical backing. Displacing an H atom, which is small, reactive and chemically bonded to the rest of the A site cation will not 'heal'. Would it not be likely that the deprotonated H will react with the interstitial I and form HI? The authors need to carry further experiments and analysis to determine the exact chemical mechanisms that occur when there is an NIEL event and the subsequent IEL event.

12. In figure 4, the efficiency actually increases after the exposure to radiation compared to the initial performance? What is the explanation for this, and how is this 'aging' effect accounted for in the analysis?

Response to Reviewers' comments on NCOMMS-22-53195

Reviewer #1 (Remarks to the Author):

The manuscript by Kirmani et al. investigates the different damage mechanism of high-energetic irradiation on the performance of perovskite solar cells. This is directly important for future space applications, and, beyond that provides an insight into healing mechanisms in halide perovskites that could be important for other aspects/fields as well. The manuscript is well written, and the experiment dedicated to isolate potential defect healing due to local heating, is very well designed. In my opinion the paper will be highly interesting for the scientific community, and I'd be happy to recommend an improved version of the paper for publication.

We thank the reviewer for appreciating the importance of these results, the experimental design, and for their very supportive comments. As per the reviewer's suggestions, we have now included experiments and discussion regarding other plausible explanations for the observed healing, and have also discussed the role charge transport layers might be playing in the observed device PCE trends. The reviewer is thanked for these helpful suggestions which we believe have improved the narrative.

Open questions:

- In the manuscript you focused on healing induced by the second irradiation with higher IEL/NIEL fraction. And while the experimental design with keeping the NIEL damage constant is extremely elegant I was wondering if you observe healing, e.g. upon thermal annealing, light illumination or gamma irradiation as well ?

Response: Motivated by the reviewer's suggestion, we designed a follow-up study to understand the impact of dark aging, light illumination, and thermal annealing on the proton-irradiated perovskite solar cells. This was also motivated by a similar question from Reviewer #3. The experimental design and results are summarized below.

Three independent experiments were designed to understand the origins of the healing mechanism. For each, three different *PIN* perovskite solar cells (PSCs) were considered: a non-irradiated device, a device exposed to the NIEL-dominant 0.06 MeV protons, and a dual dose device. The dual dosing was carried out at an optimal healing condition ($IEL \times = 1.6$). If any of these conditions were to promote healing, we would expect the single dose devices to recover in PCEs and approach the PCEs of the dual dose devices that had been healed by the 1.0 MeV proton radiation. As we now show, while light soaking was found to degrade all the cells, dark aging and thermal annealing promoted performance recovery.

The following text and supplementary figures have been added (**Page 14-15, Supplementary Fig. 15-16**):

To further probe and elucidate the working mechanism of defect healing, we designed three independent tests: dark aging, AM 1.5G light soaking, and thermal annealing. We posited that these conditions could provide the required thermal energy for rearrangement of displaced atoms.

A non-irradiated, a single dose irradiated, and a dual dose irradiated device ($IEL \times = 1.6$) were considered, and all the tests were carried out in a N₂ glove box. If any of these conditions were to promote healing, we would expect the single dose devices to recover in PCEs and approach the PCEs of the dual dose devices that have been healed by the 1.0 MeV proton radiation.

Supplementary Fig. S15 shows the effect of dark aging followed by AM 1.5G light soaking. While the single dose device (as expected) started with a lower PCE remaining factor (0.59 ± 0.10) compared to the dual dose device (0.65 ± 0.07), improvement was observed during dark aging. Over the course of 10 days in this condition, remaining factors for both devices increased to 0.77. This observation offers a key insight into the nature of radiation-induced defects in PSCs. These defects have low activation energies and undergo room temperature annealing over a sufficiently long time of 10 days. These devices were subsequently exposed to the simulated AM 1.5G solar spectrum for a total of 15 mins. Light soaking was not found to have any positive effect on the devices which showed a remaining factor drop due to light soaking and recovered to pre-light soaking values only after a day of dark aging.

Thermal conditioning was carried out on a separate set of PSCs to further understand the defect healing behavior observed for the dark aged devices. Given the tendency of these defects to heal at room temperature over multiple days, we expected this healing to accelerate and occur over a few hours at elevated temperatures. Thermal annealing data is shown in Supplementary Fig. S16 which shows JSC, VOC, FF, and PCE remaining factors as a function of annealing time for 60°C and 90°C. Annealing was carried out in a N₂ glove box and the devices were tested after 2, 5, and 12 hrs. While the PCE remaining factor of the dual dose device largely remained unchanged after 12 hrs of 60°C annealing, the single dose device showed an obvious improvement. In fact, this device recovered and achieved a similar remaining factor (~ 0.85) as the dual dose device after 5 hrs. This is clear evidence of thermally activated healing of radiation-induced defects and substantiates the observation that the IEL-dominated 1.0 MeV protons affect phonon-mediated repositioning of the atoms initially displaced by the NIEL-dominated 0.06 MeV protons leading to damage recovery.

The 90°C annealing experiments offer another key insight into thermal behavior of irradiated PSCs. The single dose devices are found to undergo significant degradation within 2 hrs of annealing at this temperature. Surprisingly, the dual dose devices show a markedly higher thermal resilience. While only a few minutes of annealing at this high temperature might be enough for healing, it appears that the initial defect density also dictates temperature stability. The reduced defect density in the dual dose devices results in a higher temperature resilience compared to the single dose devices. Suppressing radiation-induced defects can therefore prolong the lifetime of PSCs in space environments where harsh temperatures are expected. The non-irradiated devices with the least defect density do not show any performance loss after 12 hrs of annealing.

Supplementary Fig. 15. PCE remaining factors for PIN solar cells for dark aging and light soaking experiments.

Supplementary Fig. 16. Remaining factors of *PIN* solar cells for the temperature annealing experiments at 60°C and 90°C.

- The mechanism that “IEL causes phonon vibrations in the lattice sufficient for driving the displaced atoms back to their lattice position” seems plausible, however I was wondering if other explanations exist/can be ruled out: For example one could (hypothetically) argue that the higher proton irradiation interacts more with the inorganic lattice, thereby creating defects that are different in nature. Then in a second step, created defects could interact.... potentially generating defect clusters /phases that (again hypothetically) are more benign than the individual defects. Moreover, creating of PbI₂ can act as a passivation while at higher perovskite/PbI₂ ratios device performance can degrade. See e.g. Q. Chen, H. Zhou, T.-B. Song, S. Luo, Z. Hong, H.-S. Duan, L. Dou, Y. Liu, Y. Yang, *Nano Lett.* 2014, 14, 4158. Or L. Chen, J. Chen, C. Wang, H. Ren, Y. Luo, K. Shen, Y. Li, F. Song, X. Gao, J. Tang, 2021, DOI 10.1021/acsami.1c02929. or T. J. Jacobsson, J.-P. Correa-Baena, E. Halvani Anaraki, B. Philippe, S. D. Stranks, M. E. F. Bouduban, W. Tress, K. Schenk, J. Teuscher, J.-E. Moser, H. Rensmo, A. Hagfeldt, *J. Am. Chem. Soc.* 2016, 138, 10331.

Response: We thank the reviewer for offering an alternate explanation and for the references. Motivated by this question and by another concern from Reviewer 3, we carried out an in-depth literature review focusing on ion irradiation of metals and ceramics. We found that IEL-induced healing has been explained in these systems by invoking the concept of ‘localized inelastic thermal spikes’, which are sudden temperature rises as the system is pushed out of equilibrium due to IEL. As the system approaches thermal equilibrium, the extra thermal energy is transferred to the lattice via electron-phonon coupling resulting in repositioning of the displaced atoms. Finding this hypothesis from an adjacent research field, we have added a paragraph focusing and expanding on this explanation in the context of perovskites and citing the relevant references on **Page 11**.

- Especially in the nip structure the main effect in PCE variation (degradation and healing) is from the FF. Why is this the case ? Trap assisted recombination should increase non-radiative recombination, and hence one could expect also a decrease of the Voc, maybe in conjunction with a decrease in luminescence (be it photo or electro-luminescence) ... Do you observe increased non-radiative recombination/Can you exclude other transport/charge extraction issues as main cause for the drop in FF/JSC?

- Could degradation of the spiro-OMeTAD also play a role? Especially since the impact on FF is case of the nip device is quite strong? See e.g. J. Barbé, D. Hughes, Z. Wei, A. Pockett, H. K. H. Lee, K. C. Heasman, M. J. Carnie, T. M. Watson, W. C. Tsoi, *Sol. RRL* 2019, 1900219, 1.

Response: We thank the reviewer for these questions regarding the possibility of device layers other than the perovskite active layer contributing to the observed radiation-induced device degradation and healing. We take this opportunity to mention that this paper focuses on the perovskite absorber degradation and the dual dose experiments were accordingly designed to target the full device stack, representing the damage profile expected in the space environment. This was done by choosing proton energies that fully traverse the entire device stack (0.06 MeV and 1.0 MeV) creating uniform damage throughout. Results from thermal admittance spectroscopy (TAS) shown in **Figure 3A** are evidence of defects in the perovskite active layer due to atomic

displacement and subsequent formation of charged iodide interstitials upon proton irradiation and agree with previous reports where TAS has been used to pinpoint defects in the perovskite layer (Nat. Commun. 5, 5784 (2014), Nature Energy 7, 65-73 (2022)).

We agree with this point raised by the reviewer and have added the following paragraph on **Page 15-16**. We also carried out further SRIM simulations to guide future studies that can specifically target charge transport layers and interfaces, besides the active layer. This is shown in the newly added **Supplementary Fig. 17**. These are shown below as a reference.

“Finally, while this study provides direct evidence of defect creation and healing in the perovskite active layer, the possibility that the other device layers and interfaces might also be contributing to the observed radiation response cannot be ruled out. For example, the bis(trifluoromethane)sulfonimide lithium (Li-TFSI) salt used to dope the spiro-OMeTAD hole transporter for NIP PSCs is known to actively migrate across the device (<https://pubs.rsc.org/en/content/articlelanding/2017/EE/c7ee00358g#!divCitation>; <https://pubs.aip.org/aip/jap/article/89/9/4986/179740/Lithium-doping-of-semiconducting-organic-charge>). It is possible that proton irradiation displaces Li-TFSI resulting in de-doping of spiro-OMeTAD that can result in loss in FF. Future studies aiming to specifically probe the interface effects will require targeted proton irradiation by accordingly tuning the incident proton energy. Supplementary Fig. S17 shows vacancy profiles in PSCs from incident protons beams with energies of 0.01 MeV, 0.03 MeV, and 0.06 MeV. 0.01 MeV protons mostly damage the hole transporting interface, while the 0.03 MeV protons create vacancies mostly in the perovskite active layer without affecting the electron transporting interface. These targeted irradiation experiments can be used to decouple the irradiation effects on charge transport layers and the active layer.”

Supplementary Fig. 17. (Top). SRIM simulations showing proton straggling for 0.01 MeV, 0.03 MeV, and 0.06 MeV protons incident on *PIN* solar cells. (Bottom). Corresponding vacancy profiles. Shaded region highlights the perovskite active layer.

- Are there any conclusions that can be drawn from these investigations to direct future irradiation studies ?

Response: These experiments provide direct evidence of the unique radiation response of PSCs and demonstrate that radiation response of these solar cells is dependent on the incident particle energy and fluence, unlike the conventional space solar cells where radiation effects are governed only by the NIEL that defines the displacement damage dose (DDD).

We believe this investigation suggests that future studies aimed at making an assessment of PSC radiation tolerance should carry out comprehensive radiation testing using a wide range of incident particle energies and fluences relevant to the space environment.

- Lastly I have a question regarding the defect spectroscopy: Just recently Futscher et al (<https://pubs.acs.org/doi/full/10.1021/acseenergylett.1c02076>) showed that “ In mixed ionic–electronic conductors such as HaPs or ClISs, the ionic responses often mask electronic defects, regardless of whether a voltage pulse, a current pulse, or a light pulse causes the disturbance, or

whether current, capacitance, or voltage is the quantity to be observed.” and then concluded that one has to be careful that measured defect bands are due to electronic defects (excluding ionic) as otherwise erroneous conclusions may be drawn.

Response: The reviewer is thanked for this reference. As mentioned by Futscher et al in the above paper, for the ionic defects the time scales for drift and diffusion of ions are on the order of milliseconds to seconds. This would only impact the measurement of defects with TAS at the low frequency domain (Hz up to 1 kHz) and requires caution to distinguish between different defect types. In our work, the major change of defects were trap bands I and II which were measured at AC frequencies ranging from 10 kHz to 2 MHz, which was significantly higher than that of the response frequency for ionic defects. These defects have been identified as negatively and positively charged iodide interstitial defects, respectively, for iodide-based perovskite (Nature Energy 7, 65-73, doi:10.1038/s41560-021-00949-9 (2022)).

We have now added the following paragraph on **Page 9-10**:

“We note that in Fig. 3A the major changes observed are in trap bands I and II which are measured at AC frequencies in the range of 10 kHz – 2 MHz. These are significantly higher than the response frequencies for drift and diffusion due to ionic defects (up to 1 kHz) which occur on timescales of milliseconds to seconds. This implies that the defects measured here are different from ionic drift and diffusion.”

Reviewer #2 (Remarks to the Author):

This work by A. Kirmani et al. reports the degradation and healing behaviors of lead halide perovskite solar cells under energy-tuned proton radiation sources. A low-energy of 0.06 MeV and a high-energy of 1 MeV protons were used in this studied. The radiation-matter interactions, energy loss, and vacancy densities of the exposed perovskite films were studied using SRIM simulations. The device performances and defect densities were also examined experimentally. The main discovery is the radiation-induced efficiency recovery under high-energy radiations with optimal doses, which is likely due to electronic ionization, lattice vibrations and local heating. This work provides important insights on the radiation effects on perovskite solar cells for space applications. I can recommend publication after the following comments and concerns being addressed.

We thank the reviewer for their supportive and careful assessment of this work. We have carried out the key experiments suggested by the reviewer. This includes time-resolved photoluminescence (TRPL) and dual dose irradiation on higher PCE perovskite solar cells (21-22%).

1. Based on Fig S1, it seems that the difference in IEL/NIEL for 0.06 and 1 MeV is not very large. What about even lower or higher proton energies? For example, 0.01 eV vs 10 MeV?

Response: We have now added horizontal lines on **Supplementary Fig. 1** as a guide to the eye to highlight the significant difference in the IEL/NIEL ratio between 0.06 and 1 MeV protons. In fact, the ratio increases noticeably from ~600 to ~2000 between these two proton energies. This is the

reason we have used these energies for the dual dose experiments in this manuscript. Another key reason behind our choice is that proton energies in this low-energy range are the most prominent in space environments. In fact, as we have shown recently, energies below 0.05 MeV cannot create a uniform damage profile within the perovskite necessary to mimic space conditions, such low energy protons are absorbed in the topmost layers of the stack, in this case the metal electrodes.

Nevertheless, the reviewer is indeed correct that the choice of 0.01 and 10 MeV can further increase this ratio, as shown below. However, while 0.01 MeV protons will not fully penetrate the device stack (**Supplementary Fig. 17**), the 10 MeV proton fluence is significantly lower than the most abundant, low-energy protons in most space orbits of interest, as shown in panel (b) below (**Response Figure 1**). Additionally, 10 MeV protons are well beyond the energy range of our accelerator.

Response Figure 1. (a). IEL/NIEL ratio as a function of proton energy, and (b). proton fluence curves for common space orbits of interest as a function of proton energy.

2. Since the devices will be exposed to radiation only when they are used in space, I'm wondering if the experiments were carried out in a simulated space environment?

Response: The proton irradiation experiments were carried out under a vacuum of 1×10^{-7} Torr, a high vacuum environment, as we have mentioned in the Methods section (**Page 19**). While the space environment combines ultrahigh vacuum, temperature cycling and, space radiation, and the best way to realize this is to launch solar cells into the space, we believe irradiating the devices with protons most representative of space radiation (0.06 MeV and 1.0 MeV) under a high vacuum is a close and viable simulation of space environment.

3. In page 6, the authors claimed “the NIEL-dominated protons create atomic displacements and leads to non-radiative recombination”. But there is no any direct proof of increase of decrease in non-radiative recombination. Can PL and TRPL be measured?

Response: The reviewer is thanked for this question and prompted us to carry out TRPL experiments on *PIN* solar cells exposed to various irradiation conditions. **Supplementary Fig. 11** below shows TRPL data and **Supplementary Table 1** summarizes the carrier decay times and the

corresponding decay amplitudes obtained from a bi-exponential fit. These experiments were carried out on the device stack without the metal electrode. The longer decay component (τ_2) undergoes a ~50% reduction to 28.1 ns after single dose 0.06 MeV proton irradiation, compared to the non-irradiated device (53.9 ns). Dual dose irradiation corresponding to the IEL/NIEL ratio of 1.6 leads to a recovery of this decay component to 36.8 ns. Increasing the IEL/NIEL ratio to 2.4 further increases this to 46.8 ns, a value closer to the non-irradiated sample. This trend follows the change in PCE remaining factors corresponding to these irradiation conditions (Table 2), indicating that the irradiation-induced device performance changes are indeed correlated with non-radiative recombination centers.

We have now included the following paragraph on **Page 13-14** discussing this and have also added this data to the SI. These are shown below as a reference:

“Time-resolved photoluminescence (TRPL) measurements on these device stacks add support to the defect creation and healing mechanism described above. Data shown in Supplementary Fig. and Supplementary Table S highlights a 50% reduction in the longer decay component (τ_2) for the single dose irradiated stack (28.1 ns) compared to the non-irradiated sample (53.9 ns), indicating creation of non-radiative recombination centers. τ_2 increases for the dual dose samples as IEL \times is raised from 1.6 (36.8 ns) to 2.4 (46.8 ns) closer to the non-irradiated sample and pointing toward healing of the recombination centers. Taken alone, this may not prove the point because of the small differences in the lifetimes, but the trend is supportive of the hypothesis.”

Supplementary Fig. 11. TRPL plots and fitting curves on device stacks without metal electrode.

Supplementary Table 1. Carrier decay times and decay amplitudes obtained from a biexponential fit to the TRPL curves.

Device	A₁	τ_1 (ns)	A₂	τ_2 (ns)
Not irradiated	0.22	2.3 ± 0.0	0.89	53.9 ± 2.0
Single dose	0.46	3.5 ± 0.1	0.55	28.1 ± 0.6
IEL x = 1.6	0.44	3.9 ± 0.0	0.58	36.8 ± 0.8
IEL x = 2.4	0.38	3.7 ± 0.1	0.73	46.8 ± 1.3

4. Because IEL may cause lattice vibration and local heating, can the local temperature be measured and quantified?

Response: This is an interesting question and we believe can lead to precise, atomic-level insights into dynamics of defect creation and relaxation. However, with the tools available to us we cannot measure the local temperature. This can possibly be an exciting separate study involving photophysical characterization backed by molecular dynamics simulations.

5. I noticed the initial device efficiencies are quite low (16~18%). Why not using higher performance devices in this study? – many of the NREL experts can make >24% PCE.

Response: Acting on the reviewer’s suggestion, we collaboratively fabricated higher PCE *PIN* solar cells with efficiencies in the range of 21-22%. The trends for the single- and dual dose irradiations are shown in the figure and summarized in the table below (**Response Figure 2, Response Table 1**).

Response Figure 2: J - V curves of representative higher PCE PIN solar cells prior to irradiation (grey), after irradiation with the NIEL-dominated single dose of 0.06 MeV protons (cyan), and after being healed with dual dose radiation (red, magenta).

Response Table 1. Summary of higher PCE *PIN* device parameters for various irradiation conditions.

Device	Rad. 1 (MeV, cm ⁻²)	Rad. 2 (MeV, cm ⁻²)	IEL x	J_{sc} (mA.cm ⁻²)	V_{oc} (Volts)	FF	PCE (%)	PCE Remaining factor
1.	0.06, 1×10 ¹³	-	-	23.99 ± 0.31	1.13 ± 0.01	0.79 ± 0.00	21.46 ± 0.89	0.83 ± 0.08
				22.21 ± 1.31	1.13 ± 0.02	0.71 ± 0.02	17.84 ± 1.53	
2.	0.06, 0.97×10 ¹³	1.0, 1.13×10 ¹³	1.6	23.72 ± 0.35	1.13 ± 0.00	0.79 ± 0.01	21.32 ± 0.68	0.75 ± 0.06
				22.31 ± 0.83	1.13 ± 0.00	0.63 ± 0.02	15.99 ± 1.12	
3.	0.06, 0.93×10 ¹³	1.0, 2.64×10 ¹³	2.4	24.11 ± 0.21	1.13 ± 0.00	0.80 ± 0.01	21.92 ± 0.46	0.67 ± 0.04
				22.07 ± 0.60	1.10 ± 0.00	0.60 ± 0.01	14.64 ± 0.74	

The NIEL-dominated single dose irradiation (0.06 MeV protons) was unable to create the same level of damage in these higher PCE cells, resulting in higher remaining factors that are less sensitive to the higher IEL dose. These devices were fabricated using a recently reported deposition technique where the hole-selective contact was spontaneously formed during perovskite layer fabrication (Zheng, X, et al, *Nature Energy*, **2023**). These differences in fabrication method and device behavior are too substantial to compare directly. While this can be a separate in-depth study, the effect of radiation damage and healing seems to be amplified in scenarios where the NIEL-dominated first dose creates significant damage. Nonetheless, it is harder to parse the differences, but we expect that the self-healing physics in higher PCE cells will follow the mechanism outlined in this study.

6. A little more background introduction will be helpful. For example, why proton radiation is selected in this study? Compared with other radiations, is proton a major concern in space? What happens when the devices are exposed to a combination of high, mid, low protons, together with X-ray, gamma-ray, and alpha and beta rays?

Response: We have now added an explanation for our choice of proton radiation. We have referenced recent reports including our earlier publication (Kirmani et al, *Joule*, **6**, 2022) where the importance of using low-energy protons for PSC radiation testing has been elaborately discussed. The following sentences were added to **Page 2** along with **Supplementary Fig. 1**.

“Protons are particularly important for radiation testing of PSCs due to their high NIELs compared to electrons (Supplementary Fig. 1). Though alpha particles have higher NIELs, they have very low fluences in space orbits and are therefore not representative of the space environment.”

Supplementary Fig. 1. Comparison of NIELs for alpha particles, protons, and electrons for PSCs as a function of the incident particle energy.

Reviewer #3 (Remarks to the Author):

Kirmani et al show results of perovskite solar cells that have been exposed to different proton irradiation energies and dosages. Notably the study compares two of the most common device architectures in a systematic study. The authors mainly use JV characteristics to determine whether that has been a healing effect. The study however lacks any depth in terms of mechanistic insights in the physical and chemical processes that occur in the devices before and after radiation. There is also a lack of materials characterization of the perovskite material itself. A recent study by Durant et al (<https://pubs.acs.org/doi/full/10.1021/acsenerylett.1c00756>) is a similar study in terms of tuning the proton energy and fluence and comparison of JV characteristics. Moreover, there is no indication on what step researchers need to take to manipulate and enhance the self healing mechanism to fully reproducibly recover the power conversion efficiency. Because of the lack of insight into any degradation and healing processes that are occurring in the perovskite absorber itself, this manuscript as it is, does not meet the requirements, in terms of broad appeal and of significance to the field, to be published in Nature Communications. Below are more detailed comments and questions regarding the data and its interpretation.

The reviewer is thanked for their assessment. We believe their suggestions were very helpful in polishing the narrative and strengthening the conclusions. We have carried out several experiments suggested by the reviewer including time-resolved photoluminescence (TRPL) and dual dose irradiation on SiOx-capped perovskite solar cells. We have also carried out an in-depth literature review and included explanation on the defect creation and healing mechanism. We hope the revised manuscript has addressed most of the reviewer's concerns.

1. The level of detail of processes that are induced by the inelastic ionizing energy loss (IEL) mechanism are lacking. Considering the manuscript and the results are predominately attributed to this mechanism a much in-depth literature review needs to be carried out.

Response: We agree with the reviewer that more information regarding the exact role of IEL will be helpful. However, this is one of the first manuscripts to directly probe the impact of IEL on perovskites, and an understanding has not yet been established as to how exactly IEL heals the radiation-induced defects. Motivated by the reviewer's suggestion, we have now included the following discussion with references on **Page 11**:

“In organic semiconductors, electronic ionization has been shown to break C-H bonds and abstract H atoms resulting in defect creation. Thermally activated H migration results in healing of these defects. The processes of atomic displacement (NIEL) and electronic ionization (IEL) are correlated and together impact the irradiated lattice. In a timespan of femto-picoseconds following irradiation, IEL raises the electronic temperature above that of the lattice causing localized inelastic thermal spikes pushing the system far from equilibrium. Over time, as the system approaches thermal equilibrium, the electron transfers thermal energy to the lattice due to electron-phonon coupling, providing enough energy to the displaced atoms to reorganize. Impact of IEL on perovskite semiconductors can likely be explained via a similar mechanism given the presence of C-H and C-N species at the A-site.”

While the community continues to work to understand these effects, we postulate that this interplay between NIEL and IEL and the strong electron-phonon coupling in perovskites causes these devices to heal.

2. Following on from the previous point, if indeed the healing mechanism is associated with the warming of the lattice through phonons, then the authors must show the effect of annealing the device at a set temperature before and after proton irradiation can also help the healing process. Heat can also be dissipated into the lattice from hot carriers. To validate that it is indeed phonon mediated heating, the authors could excite the films with blue light after proton irradiation. The carrier cooling mechanisms, via phonons can also increase the lattice temperature.

Response: Based on these suggestions, we designed a comprehensive set of experiments to probe the effects of dark aging, light soaking, and thermal annealing on the devices. This discussion has been added to **Page 14-15** and to the **Supplementary Fig. 15-16**.

In brief, we found that while light soaking reduced the device PCEs, both dark aging (under N₂) and thermal annealing (under N₂) showed PCE improvements. Dark aging was found to improve the PCEs over a longer time period of several days, while annealing at 60°C resulted in recovery of the single dose irradiated solar cells over a few hours. This was a very helpful suggestion by the reviewer and helped us conclude that the IEL-induced healing phenomenon is indeed thermally activated.

3. What is the actual relationship between the fluence and NIELs and IELs? There is no clear evidence that increasing the fluence by a linear amount would result in the same proportionate increase in NIELS over IELS. This assumption seems arbitrary, without any scientific basis.

Response: In this paragraph we aim to explain this relationship and the rationale behind our choice of the fluences. Any charged particle passing through matter loses energy via NIEL and IEL. The ratio between the two is defined by a highly complex set of quantum mechanical equations that form the basis of scattering theory. This mathematical formalism is built into the SRIM code that has been used to guide the choice of proton energies and fluences. SRIM is a well-established platform routinely used in nuclear physics to simulate interactions of incident ions with matter.

Figure 1C-D copied below is the outcome of these SRIM simulations on an *NIP* device and shows the NIEL and IEL profiles in the device for various proton energies and their dual dose combinations. As can be seen, the mathematical model predicted that IEL is higher than the NIEL.

Figure 1C-D: NIEL and IEL profiles for the various irradiation scenarios considered are shown in C and D. By selecting the two radiation conditions (energy and fluence) shown with dashed lines in C, the NIEL created in each cell is the same, whereas the IEL increases up to factor of 27.5 greater for increasingly higher energy dosing as shown in D. This enables us to tune the IEL/NIEL ratio.

A linear increase in the fluence will result in a linear increase in the total NIEL and IEL, however, their ratio will remain constant. To tune this ratio, the incident proton energy needs to be tuned, as highlighted by **Supplementary Fig. 2** which is copied below as a convenience to the reviewer.

Supplementary Fig. 2. IEL/NIEL ratio as a function of proton energy calculated using SR-NIEL.1,2 Horizontal gray lines highlight that the ratio increases from ~600 to ~2000 from 0.06 MeV to 1.00 MeV proton energy.

Supplementary Fig. 2 highlights that a proton of higher energy is less likely to cause atomic displacement via NIEL and is more likely to create electronic ionization. We hope that the relationship between NIEL and IEL is now clearer to the reviewer.

4. On page 6 the authors mention ‘Importantly, this platform can be used to explore an IEL-related phenomenon, such as the self-healing, without changing the vacancy profile in the PSCs incurred from NIEL.’ What is the vacancy profile induced by NIELs? Please may the author back this statement with a literature review.

Response: As protons pass through the device stack, the deposited NIEL causes atomic displacements and vacancy creation. We have shown both the NIEL profile and the resulting vacancy profile in Figure 1, however, to simplify this point we have replotted this now as below (**Response Figure 3**). As can be seen, the vacancy profile has the same shape as the NIEL profile. We hope this point is now clearer.

Response Figure 3. NIEL (top) and vacancy (bottom) profiles across an *NIP* device.

5. The authors have recently published a paper on SiO₂ encapsulation for radiation protection of perovskite devices (<https://www.nature.com/articles/s41560-022-01189-1>). Therefore, there should be experimental evidence and comments on which proton energy beam would be required to replicate the damage of the radiation in the perovskite devices with encapsulation. This is especially important because and perovskite device will not be employed into space without encapsulation, therefore the trajectory and proton energy will be different with and without encapsulation.

Response: We thank the reviewer for this point. Though this is a complicated experiment since the device architecture, proton energies, and the damage profile change, we went ahead and carried out the experiment backed by SRIM simulations.

We began by carrying out SRIM simulations to understand as to what proton energies would be adequate for the SiO_x cells to create the damage profiles created in a bare cell without SiO_x. Based on our previous papers (Kirmani et al, *Joule*, **6**, 2022; Kirmani et al, *Nat Energy*, **8**, 2023), 1 μm SiO_x will absorb 0.105 MeV energy from the incident proton. In other words, the incident energies of 0.06 MeV and 1.00 MeV should be adequately increased to 0.165 MeV and 1.105 MeV for

carrying out these experiments on SiO_x cells. However, our accelerator cannot generate protons in the energy range of 0.1 – 0.3 MeV. We therefore sought a way to use 0.3 MeV protons for the single dose irradiation.

Response Figure 4: Vacancy profiles in SiO_x PSCs for 300 keV protons incident at various angles simulated using SRIM. The control case of 60 keV protons normally incident on a bare cell is also shown.

It is clear from the above **Response Figure 4** that the vacancy profiles for normally-incident 0.06 MeV and 0.3 MeV protons are vastly different. Tuning the incident angle for 0.3 MeV protons reveals that 60° incidence of these protons on the SiO_x device can result in a vacancy profile similar to that created by 0.06 MeV protons normally incident on a bare device.

Response Figure 5: Radiation response of proton-irradiated $1 \mu\text{m-SiO}_x$ PIN PSCs. A. J - V curves of representative solar cells prior to irradiation (grey), after irradiation with the NIEL-dominated single dose of 0.06 MeV protons (cyan), and after dual dose irradiation (red, magenta). Remaining factors for B. J_{sc} , C. V_{oc} , D. FF , and E. PCE for the irradiated PSCs for IEL $x = 1.6$, 2.4 scenarios. The single dose control device is denoted by the IEL $x = 1.0$ case. Parameters are averaged over 5 devices.

Response Figure 5 show J - V curves and remaining factors for solar cells exposed to the NIEL-dominated single dose condition of 0.3 MeV (60°) and the dual dose conditions involving 0.3 MeV protons (60°) and 1.105 MeV protons. The single dose irradiation (0.3 MeV protons) was not able to create sufficient damage in these protected cells, resulting in high remaining factors approaching 0.90. Given a significantly low radiation-induced defect density, healing was not observed for the dual dose irradiation conditions used.

The SiO_x barrier is expected to modify radiation interaction in these cells, as discussed previously (Kirmani et al, *Nature Energy*, 2023). It is also possible that the slight differences in the vacancy profiles between the 0.3 MeV (60°) protons on the SiO_x cell and normally incident 0.06 MeV protons on a bare cell prohibited observation of the healing effect. This highlights the complications associated with the SiO_x cell irradiation, while also demonstrating the radiation protection benefits of SiO_x.

6. Whilst the study is based on perovskite devices, there is very little characterization of the perovskite thin-film itself. The manuscript extensively talks about self-healing of the perovskite thin-film and attribute IEL to just the perovskite material, however the device is made up of many materials and interfaces, and therefore the authors need to investigate to find out exactly which layer and or interface is being damaged and or ‘self-healed’.

Response: We agree with the reviewer that developing an understanding of the role of interfaces on the observed device performance trends is important. In fact, we cannot rule out the impact interfaces have on the defect creation and healing observed in this study. However, we believe this warrants a separate, comprehensive study and is beyond the scope of this current paper. We have added a paragraph discussing this on **Page 15-16** and new SRIM simulations have also been added (**Supplementary Fig. 17**).

In this study, thermal admittance spectroscopy (TAS) experiments provide direct evidence of defects created in the perovskite active layer and suggest these to be charged iodide species, in agreement with previous reports. This is in agreement with the findings from SRIM simulations. The latest TRPL experiments (**Page 13-14, Supplementary Fig. 11, Supplementary Table 1**) also point toward non-radiative recombination centers created by the single dose irradiation and healed by the IEL-dominant dual dose irradiation.

7. The authors state ‘Expectedly, the NIEL-dominated protons create atomic displacements and leads to non-radiative recombination.’ There is no experimental evidence in the manuscript that show there is an increase in non-radiative recombination. Please may the authors provide data to back this assumption up. For example, time-resolved photoluminescence traces.

Response: We thank the reviewer for this suggestion. We carried out TRPL experiments and this data is now included in the SI along with a discussion (**Page 13-14, Supplementary Fig. 11, Supplementary Table 1**).

In brief, we found that the NIEL-dominated single dose irradiation resulted in a 50% decrease in the carrier decay time pointing toward non-radiative recombination. Dual dose irradiation healed these defects increasing the decay time similar to the non-irradiated films.

8. Please state how many devices were included for all the data and for the error analysis. For example, in Supplementary Figure 5.

Response: We have now included this information to all the relevant figures. The reviewer is thanked for pointing this.

9. For the NIP device the observed trend is not statistically significant. Why is this? Majority of the changes that occur in the device are from the Fill Factor. Why is this assigned to the suppression of non-radiative recombination? Usually, a change in recombination is mostly exhibited in a voltage loss. What about changes at the interface?

Response: We do not exactly know why the FF changes and not the Voc for the *NIP* devices. TAS gives direct evidence of changes happening in the perovskite layer by tracking the charged iodide defects (trap bands I and II), and is in agreement with prior reports. We now have further evidence of this in the form of TRPL.

However, we cannot rule out the effect of interfaces on the radiation induced PCE loss and healing. This is an important question and should be the focus of an independent study. It is possible that irradiation of the *NIP* devices also changes the doping density in the spiro-OMeTAD HTL resulting in FF loss.

We have now included a paragraph on **Page 15-16** discussing this possibility.

10. Please may the authors expand and give a description of what the potential mechanism is for the healing. From what is implied it seems that an atom is displaced from a NIELs event, followed by a lattice heating allowing that atom to move back. If this is the case then is the assumption that only one type of atom is displaced? If so then is IEL providing sufficient energy for only these ‘types’ of atoms to move back into the vacancy?

Response: Although the exact ‘type’ of atoms that have moved back upon healing cannot be discerned from the experiments carried out in this study, we have now included discussion regarding a potential mechanism on **Page 10** motivated by the reviewer’s comments. The reviewer is correct that based on what we understand so far about this system, the radiation damage and recovery in perovskites appears to be initiated by NIEL followed by IEL-induced lattice heating allowing the displaced atoms to move back.

11. From the tDAS data and the SRIM analysis the authors infer that H and I atoms are expected to be displaced most. The following explanation on the healing has very little chemical backing. Displacing an H atom, which is small, reactive and chemically bonded to the rest of the A site cation will not ‘heal’. Would it not be likely that the deprotonated H will react with the interstitial I and form HI? The authors need to carry further experiments and analysis to determine the exact chemical mechanisms that occur when there is an NIEL event and the subsequent IEL event.

Response: We wish to point out that the SRIM simulations do not take into account the chemical bond strengths. The role bond strengths play in defining radiation effects remains to be understood. Therefore, while more work is needed to get a clearer picture of the defect creation and healing phenomenon in perovskites, the experiments we have designed induce, tune, and probe these processes and, together with theoretical modeling, present the best preliminary picture to date of the mechanisms involved.

In organic semiconductors, precedent exists for H atoms being abstracted from the C-H bonds and getting displaced, followed by them moving back upon annealing resulting in healing. (Street et al, Phys. Rev. B., **85**, 2012, <https://journals.aps.org/prb/abstract/10.1103/PhysRevB.85.205211#fulltext>). We have postulated that a similar mechanism might exist in perovskites given the C-H and C-N species at the A-site. Further, the evidence for charged iodide species forming defects is provided by our TAS experiments.

The following paragraph has now been added to **Page 11** to reflect this:

“We note that SRIM simulations do not consider chemical bond strengths and the role bond strengths play in defining radiation effects remains to be understood. Therefore, while more work is needed to get a clearer picture of the defect creation and healing phenomenon in perovskites, the experiments we have designed induce and probe these processes and, together with theoretical modeling, present the best preliminary picture to date of the mechanisms involved.”

12. In figure 4, the efficiency actually increases after the exposure to radiation compared to the initial performance? What is the explanation for this, and how is this 'aging' effect accounted for in the analysis?

Response: We apologize for not being able to fully understand this question. Figure 4 shows the PCE remaining factors for the various IEL/NIEL scenarios as this ratio is tuned by increasing the 1.0 MeV proton fluence. In order for the device efficiency to increase above the initial performance, the PCE remaining factor needs to be above 1.0. However, in Figure 4, the PCE remaining factor never reaches 1.0. Though there is an improvement in PCEs after the initial radiation-induced damage by 0.06 MeV protons, it never improves more than that the initial PCE. Also, we wish to clarify that Figure 4 does not show aging data for the devices. All devices were measured within a couple days of proton irradiation, as soon as they were received from the radiation facility. For the reviewer’s convenience, we have copied Figure 4 below.

Fig. 4. Radiation response of proton-irradiated PIN PSCs. A. J - V curves of representative PIN solar cells prior to irradiation (grey), after irradiation with the NIEL-dominated single dose of 0.06 MeV protons (cyan), and after being healed with dual dose radiation (magenta). Remaining factors for B. J_{sc} , C. V_{oc} , D. FF , and E. PCE for irradiated PIN PSCs for various IEL x scenarios. The single dose control device is denoted by the IEL $x = 1.0$ case. Black arrow indicates IEL-induced healing and subsequent increase in PCE for the IEL $x = 2.4$ case.

REVIEWER COMMENTS

Reviewer #1 (Remarks to the Author):

Additional Experiments performed by Kirmai et al, not only answered all my previous questions but also improved the understanding and thus impact of the manuscript significantly.

It is a pleasure to recommend the manuscript now for publication without further revision.

Reviewer #2 (Remarks to the Author):

The authors have addressed reviewers' questions and concerns carefully. In my opinion this work can be accepted now.

Reviewer #3 (Remarks to the Author):

Kirmani et al have attempted to explain the points that were addressed to them. Whilst there has been more investigations and experiments, the results do not convincingly show the generality of the mechanisms and insights they carried out in the initial investigations. The Response Figure 2 and Response Figure 5, where they have repeated the study with high performing solar cells do not show the trend that they observe in the initial investigations, indicating that the initial performance of the device is critical in the healing affect. There may be many factors that may influence healing in a 16-17% PCE devices however when the trap density is low and the interface quality is high as in the >20% devices there does not seem to be a healing affect. This alone shows that the insights shown are very specific and cannot be used to form a foundation of understanding on radiation induced healing. Regarding the statement, 'These differences in fabrication method and device behavior are too substantial to compare directly.' the technique is still solution processing and the end product is a perovskite thin film, additionally this goes against the idea that the mechanism presented applies to all perovskite devices.

Moreover, there is not a substantial explanation given about the increase in fill factor (it was the most influenced and therefore needs to be looked at much more closely, yet the discussion is based on trap density which is more representative of the voltage).

The authors misunderstood the comment regarding annealing. To clarify both experiments of thermal annealing and illumination with a blue light source (to produce hot carriers that will then cool) need to be carried out to show indeed that it's a hot carrier induced healing and not just general thermal

annealing that 'heals' the device. This may give credit to the idea that its electron-phonon based healing that the authors propose.

Overall, the new additions and changes to the manuscript have not clarified the initial results and explanations, there is now a contradiction that the proposed mechanism is not general and seems to only work in lower performing devices. Therefore, I cannot recommend the paper for publication nature communications.

Response to Reviewers

The 3 reviewers are also thanked for their overall positive assessment and for strengthening the manuscript via constructive suggestions.

We are excited to learn that Reviewers 1 & 2 are now satisfied with the revision and, considering Reviewer 3's concerns, have offered further minor suggestions to reach a more informed and balanced decision.

Reviewer 1: *“Additional Experiments performed by Kirmani et al, not only answered all my previous questions but also improved the understanding and thus impact of the manuscript significantly. It is a pleasure to recommend the manuscript now for publication without further revision.*

Reviewer 2: *“The authors have addressed reviewers' questions and concerns carefully. In my opinion this work can be accepted now.”*

Reviewer 3: *“Kirmani et al have attempted to explain the points that were addressed to them. Whilst there has been more investigations and experiments, the results do not convincingly show the generality of the mechanisms and insights they carried out in the initial investigations. The Response Figure 2 and Response Figure 5, where they have repeated the study with high performing solar cells do not show the trend that they observe in the initial investigations, indicating that the initial performance of the device is critical in the healing affect. There may be many factors that may influence healing in a 16-17% PCE devices however when the trap density is low and the interface quality is high as in the >20% devices there does not seem to be a healing affect. This alone shows that the insights shown are very specific and cannot be used to form a foundation of understanding on radiation induced healing. Regarding the statement, ‘These differences in fabrication method and device behavior are too substantial to compare directly.’ the technique is still solution processing and the end product is a perovskite thin film, additionally this goes against the idea that the mechanism presented applies to all perovskite devices.*

Moreover, there is not a substantial explanation given about the increase in fill factor (it was the most influenced and therefore needs to be looked at much more closely, yet the discussion is based on trap density which is more representative of the voltage).

The authors misunderstood the comment regarding annealing. To clarify both experiments of thermal annealing and illumination with a blue light source (to produce hot carriers that will then cool) need to be carried out to show indeed that it's a hot carrier induced healing and not just general thermal annealing that 'heals' the device. This may give credit to the idea that its eletron-phonon based healing that the authors propose.

Overall, the new additions and changes to the manuscript have not clarified the initial results and explanations, there is now a contradiction that the proposed mechanism is not general and seems to only work in lower performing devices. Therefore, I cannot recommend the paper for publication nature communications.”

Based on the editor's recommendation informed by confidential remarks from the Reviewers 1 & 2, as communicated to us:

“Reviewer #1 notes that the contradictory results (Response Figure 2 and Response Table 1) should not be neglected but to show them in the supplementary information of the paper, with discussion in the main text explicitly to mention that the healing is not observed in higher performing and more

stable devices, which requires further investigation. Reviewer #1 also questions if the observed fill factor healing is related to the different hole transport layer used. Meanwhile, Reviewer #2 also suggests that it will be helpful to add related discussions with explanations on the fill factor."

we here offer a more complete revised version of the manuscript. Here is a summary of the changes made in this revision addressing the reviewers' latest comments:

- a) The additional data that were previously only included in the Response Letter have now been included as Supplementary Information (Supplementary Figure 18-20, Supplementary Table 2, Supplementary Note 1-2). A discussion has been included in the main text elaborating on these new data (Page 16).
- b) A hypothesis has now been included in the main text regarding the observed healing of fill factor for the NIP solar cells (Page 16).